# An improved long short term memory network for intrusion detection

**Asmaa Ahmed Awad[2], Ahmed Fouad Ali[2,3☯], Tarek Gaber** [iD][1,2☯] *

**1** School of Science, Engineering and Environment University Salford, Manchester, United Kingdom,
**2** Department of Computer Science, Faculty of Computers and Informatics, Suez Canal University, Ismailia,
Egypt, **3** Faculty of Information Technology, Misr University for Science and Technology, Egypt

☯ These authors contributed equally to this work.
* t.m.a.gaber@salford.ac.uk

improved long short term memory network for
intrusion detection. PLoS ONE 18(8): e0284795.

SERBIA

**Data Availability Statement:** The data underlying
the results presented in the study are available
from (http://nsl.cs.unb.ca/NSL-KDD/). The
standard dataset, NSL-KDD 2009 (Network
Security Laboratory- Knowledge Discovery and
Data Mining) was used and it is available at https://

## Abstract

Over the years, intrusion detection system has played a crucial role in network security by
discovering attacks from network traffics and generating an alarm signal to be sent to the
security team. Machine learning methods, e.g., Support Vector Machine, K Nearest Neigh-
bour, have been used in building intrusion detection systems but such systems still suffer
from low accuracy and high false alarm rate. Deep learning models (e.g., Long Short-Term
Memory, LSTM) have been employed in designing intrusion detection systems to address
this issue. However, LSTM needs a high number of iterations to achieve high performance.
In this paper, a novel, and improved version of the Long Short-Term Memory (ILSTM) algo-
rithm was proposed. The ILSTM is based on the novel integration of the chaotic butterfly
optimization algorithm (CBOA) and particle swarm optimization (PSO) to improve the accu-
racy of the LSTM algorithm. The ILSTM was then used to build an efficient intrusion detec-
tion system for binary and multi-class classification cases. The proposed algorithm has two
phases: phase one involves training a conventional LSTM network to get initial weights, and
phase two involves using the hybrid swarm algorithms, CBOA and PSO, to optimize the
weights of LSTM to improve the accuracy. The performance of ILSTM and the intrusion
detection system were evaluated using two public datasets (NSL-KDD dataset and LITNET-
2020) under nine performance metrics. The results showed that the proposed ILSTM algo-
rithm outperformed the original LSTM and other related deep-learning algorithms regarding
accuracy and precision. The ILSTM achieved an accuracy of 93.09% and a precision of
96.86% while LSTM gave an accuracy of 82.74% and a precision of 76.49%. Also, the
ILSTM performed better than LSTM in both datasets. In addition, the statistical analysis
showed that ILSTM is more statistically significant than LSTM. Further, the proposed ISTLM
gave better results of multiclassification of intrusion types such as DoS, Prob, and U2R
attacks.

## 1 Introduction

With the growth of the internet and the increasing use of technology in our daily lives, cyber-
crime has become a major concern for individuals, businesses, and governments alike.

www.unb.ca/cic/datasets/nsl.html. A second dataset was also used and it is available here: https://dataset.litnet.lt/data.php.

**Funding:** The authors received no specific funding for this work.

**Competing interests:** The authors have declared that no competing interests exist.

Cybercrime refers to criminal activities that are carried out using computers or the internet, such as hacking, phishing, identity theft, and malware attacks [1]. Many of the applications (like online banking, e-commerce, and healthcare services) which we use in our daily lives contain confidential and personal information that needs to be protected. To protect these applications, it is important to take a proactive approach to cybersecurity [2]. Furthermore, with the increasing number of connected smart devices in the IoT environment, there are also increasing security threats and vulnerabilities. Therefore, additional security considerations are necessary to safeguard these devices and the data they transmit. AI-based security solutions such as anomaly and intrusion detection and network traffic monitoring can be useful tools in enhancing IoT security [3, 4].

The concept of intrusion detection (ID) dates back to 1970 when it was extensively adopted to protect computer networks against both known and unknown attacks [5]. An intrusion detection system (IDS) is software that monitors a network for malicious activities and generates an alarm signal to be sent to the security team. Anomaly- and signature-based are the two main methods used in IDS. Signature-based IDS detects attacks based on matching input data with the signatures of known attacks. An anomaly-based IDS catches attacks by comparing abnormal behaviour to normal behaviour. Signature-based detection is unable to detect attacks that have not been seen before while anomaly-based detection often has high false positive rates [6].

Machine learning (ML) algorithms have been used for over 20 years to improve the performance of IDS [7]. Two types of ML have been for building anomaly detection models: shallow learning and deep learning. In general, shallow learning (Bayesian networks, support vector machines (SVMs), and artificial neural networks (ANNs)) depends on extracting features creating the prediction model [2] while deep learning has the ability to generate superior models by extracting better representations from the raw data [8]. Deep learning is a type of ML that uses artificial neural networks with multiple layers to learn hierarchical representations of data. DL can learn feature hierarchies based on massive amounts of unlabeled information, making it particularly useful for processing complex, high-dimensional data. Examples of DL algorithms include deep neural networks (DNNs) [9], convolutional neural networks (CNNs) [10], and recurrent neural networks (RNNs). One of the advantages of DL is that it can automatically learn relevant features from data, without the need for explicit feature engineering. This makes DL models more adaptable and flexible, as they can handle a wide range of input types and sizes. In the domain of intrusion detection, the most recent papers are using DL [8, 11–13]. RNN is one of the most popular deep learning algorithms for the classification of sequential data due to its recurrent (circular) manner of connections. Thus, RNN can recall all previous knowledge acquired from previous inputs during a training phase [14]. Recently, Long short-term memory (LSTM) has gained much attention due to its ability to solve the drawback of RNN in vanishing gradients by using a gating mechanism to learn long-term dependencies [15]. LSTM is also employed in attack detection, with the ability to detect unique attacks, as in [16].

Swarm intelligence (SI) algorithms (such as Butterfly optimization algorithm (BOA) [17], grey wolf optimizer [18], and particle swarm optimization (PSO) [19]) are widely used in global optimization and parameters tuning. BOA is inspired by the foraging behaviour of butterflies and it has the ability to find the optima in the hyper-search space [17]. Utilizing chaotic maps, a new version of BOA called chaotic Butterfly optimization algorithm (CBOA) has been proposed in [20]. This CBOA algorithm showed to be better than BOA in improving classification accuracy and reducing classification errors.

LSTM algorithm has been used in proposing different intrusion detection methods such as in [21] and in [22]. However, LSTM performance is impacted by extra problems with

random weight initialization [23] and overfitting [24]. In other words, although LSTM has been used in many intrusion detection systems but it still suffers from two main limitations: (1) taking high numbers of iterations to find the best weight value of its network which affects the computational costs, and (2) its classification performance is still not high. The objective of this paper is to minimize the number of iterations needed to find the best weight values of LSTM network and improving the classification performance in intrusion detection systems.

To achieve this objective, an improved version of LSTM (i.e., ILSTM) was suggested. In the ILSTM, hybrid swarm algorithms, CBOA and PSO, were employed to optimize LSTM weights while using a fewer number of iterations. The ILSTM was then used for proposing an efficient and accurate intrusion detection system for two cases: binary (normal or abnormal) and multi-class (classifying many attacks) classification.

The contribution of this work can be summarised as follows:

1. Proposing a novel and improved version of LSTM called ILSTM in which hybrid swarm intelligence algorithms (i.e., CBOA and PSO) were employed to optimize the weights of the LSTM algorithm which led to better performance using a fewer iterations.

2. Building an efficient (i.e., fewer iterations) and accurate ILSTM-based intrusion detection system for binary (normal and abnormal) and multi-class classification (classifying more than attacks such as DoS, Prob, and U2R attacks).

3. Evaluating the performance of the new ILSTM and the intrusion detection system. Thorough evaluation was done using nine performance metrics (accuracy, detection rate, false alarm rate, precision, f–measure, false negative rate, mathew correlation coefficient and kappa coefficient) under two public datasets (NSL-KDD dataset and LITNET-2020). The ILSTM performed better than LSTM in both datasets.

4. Comparing the results of the proposed solution with various deep learning algorithms. The comparison demonstrated that the proposed ISTLM gave better results in binary and multi-classification of intrusion types such as DoS, Prob, and U2R attacks.

5. Conducing statistical analysis using Wilcoxon Signed-Rank test which showed that ILSTM is more statistically significant than LSTM.

The subsequent sections of this paper are as follows: Section 2 of the paper discusses some related works on swarm intelligence, deep learning, and network intrusion detection methods. Section reference 3 contains all implemented algorithms that were used in the development of the proposed algorithm. Section 4 includes the proposed algorithm (ILSTM). Section 5 provides an experimental setup for implementation the proposed algorithm, parameter setting, performance metrics, and preprocessing phase on the NSL-KDD and the LITNET-2020 datasets. Section 6 illustrates and discusses the performance of the proposed algorithm in binary and multi-class classification as well as comparisons with other deep learning and machine learning algorithms. Section 7 presents the conclusion of this work and future work.

## 2 Literature review

IDS are critical components of computer network defense. In prior studies, several approaches proposed intrusion detection based on deep learning. In [14], RNN classifier is proposed by using one hidden layer with eighty hidden nodes and 0.1 learning rate for binary and multi-class classification on the NSL-KDD dataset. However, applying RNN has the drawback of exploding and vanishing gradients, which this method does not solve.

In [8], an integrated intrusion detection model based on a staked denoising auto-encoder and deep belief network (SADE-ELM and DBN-SoftMax)is developed to overcome the short-comings of existing deep neural network models, including their long learning times and poor classification accuracy, The proposed model only achieves 76.64% for accuracy in binary classification on the NSLKDD dataset.

The authors of [13] developed an intrusion detection model based on bidirectional long short-term memory (BiDLSTM) and convolution LSTM, and the results show that the proposed BiDLSTM is more effective than convolution LSTM. The accuracy of convolution LSTM is 89.81% but BiDLSTM reach to 94.26% in binary classification. Despite BidLSTM gives best result than convolution LSTM, it requires more training time than other compared algorithm.

The authors of [25] proposed a BAT-MC hybrid method of BLSTM and attention mechanism and compare it to other machine learning algorithms (J48, Naive Bay, NBTree, Random Forest, and SVM) using the NSL-KDD dataset in binary classification. The proposed method accomplishes 84.25% for accuracy in binary classification but has the lowest accuracy for U2R and R2L attacks in multi-class classification.

Jiang et al. [26] combined hybrid sampling techniques with deep learning networks (CNN) as a method for intrusion detection. They use one-side-selection (OSS) to reduce the noise samples in the majority categories and increase the minority categories by the synthetic minority over-sampling technique (SMOTE). The accuracy of this method is only 83.58% on the NSL-KDD dataset in binary classification and 82.74% in multi-class classification. However, while this method has a high detection rate for U2R attacks, it has a lower detection rate for other attacks such as (normal, Dos, Prob, R2L).

Chora and Pawlicki [27] studied ANN hyperparameters (activation, optimizers, batch size, epochs, layers, and neurons) for an intrusion detection model using NSL-KDD and CICIDS 2017. When using the parameters tanh, Adam, with 100, 300, 1, and 25, the accuracy was 99.9%. For the other parameters, accuracy dropped to 5.64 percent, demonstrating that the ANN model is sensitive to parameter values. They further did not consider multiclassification of intrusion types such as Dos, Prob, U2R, or R2L.

Multiple researchers have studied the use of swarm intelligence algorithms for machine learning algorithms. ELHasnony et.al [28] developed a hybrid swarm algorithm of BOA and PSO for selecting the best features. Selected features are applied for machine learning algorithm (KNN) with 5 K fold cross-validation for classification. 25 Datasets from UCI machine learning repository and COVID-19 dataset are used to evaluate the proposed algorithm, where proposed algorithm give better result than other swarm algorithms such as BOA, PSO, and GWO. ALsaleh et al. [29] investigated the impact of the salp sarm algorithm (SSA) for feature minimization on improving machine learning network-based anomaly detection classifiers such as XG Boost and Naive Bayes. Improved firefly algorithm is also proposed for optimizing parameters of XGBoost classifier for intrusion detection in [30], the proposed algorithm is tested on the NSL-KDD and UNSW-NB15 datasets. Firefly algorithm reduced the number of features to 19 from 42, where accuracy in binary classification is increased after selection features but other performance metrics are decreased, such as (precision and f-score). In multi-classification, most performance metrics give the best results after selection.

The use of swarm algorithms for deep learning networks was also investigated by researchers. As in [31], where the hybrid deep learning model CNN-OLSTM is used to detect DDos attacks and the grey wolf optimization method is present to choose the best features for detection, but it obtains a very low specificity of 51%. In [11], a feature reduction model based on correlation and information gain, followed by using a RNN classifier for the detection of

attacks and non-attacks in a reduced-feature dataset, where 90% of the NSLKDD dataset is used for training. In [32] suggested that using the whale algorithm to optimize the weights of LSTM networks to develop an effective model is called WILS, the abbreviation for whale integrated long short term memory to detect a variety of threats on IoT networks. They used the same dataset for training and testing, using 70% of the NSL-KDD as training data and the remaining 30% for testing data in binary classification.

Some research papers use mathematics algorithms for optimizing weights of LSTM, such as [33] which uses four different optimizer (metaheuristic algorithms) such as harmony search (HS), grey wolf optimizer (GWO), sine cosine (SCA), and ant lion optimization algorithms (ALOA) to train LSTM for maximizing classification accuracy.

The authors in [34] developed a model (OCNN-HMLSTM) by using lion swarm optimization (LSO) for optimization hyperparameters of CNN (spatial features) and using HMLSTM for learning temporal features. The proposed model for NSL-KDD has a binary classification accuracy of 90%, while all attack types (Dos, U2R, Prob, and R2L) have higher false positive rates, reaching 9.92%. In the research paper [35], The authors proposed the firefly algorithm for feature selection of NSL-KDD and KDD Cup 99 datasets, then used DNN for the classification process. Despite the efficiency of the hybrid eFA-DNN framework, it is only proposed for binary classification algorithms.

The authors applied an evolutionary sparse convolution network (ESCNN) in [36] for identifying and tracking attacks in distributed denial of service (DDOS) in the IoT. A variety of DDoS attack-related feature analyses were used to design the technique to reduce network overhead. The proposed network achieves a 98.28% detection rate and 99.29% accuracy in binary classification. In [37], a new feature selection strategy has been proposed using bio-inspired algorithm GWO, in addition authors applied classification method (ELM) refer to extreme learning machine. Modified GWO was tested using the UNSW NB-15 dataset and achieved 78% accuracy. In order to boost the accuracy of a machine learning classifier for intrusion detection systems, relevant features from the UNSW-NB15 and CICIDS-2017 datasets are selected using the artificial bee colony (ABC) algorithm as described in [38]. According to [39], the Firefly algorithm is also used in network intrusion detection to choose features. The Firefly algorithm can choose 10 crucial features from the KDD CUP 99 dataset, which is applied to bayesian networks (BN) and C4.5 based classifiers for anomaly detection. Image recognition has also been applied lately in intrusion detection, as in [40] where a new approach has been proposed using multistage deep learning image recognition that transforms network features into four channel images (Red, Green, Blue, and Alpha) that are used in classification. Results reach 99.8% accuracy for the BOUN Ddos dataset.

From the above literature analysis and summarized in in Table 1, it could be concluded that the performance of the deep learning-based intrusion detection system could be still improved. Such improvement should cover two aspects: binary and multi-class classifications of attacks. It was also noticed that although LSTM has been used in many intrusion detection systems, such as [21, 22, 25] but it still suffers from two main limitations: (1) taking high numbers of iterations to find the best weight value of its network which affects the computational costs, and (2) its classification performance is still not high. In addition, LSTM performance is impacted by extra problems with random weight initialization [23].

## 3 Preliminary work

In this section, an overview of the algorithms used in our proposed algorithm and intrusion detection system is given.

**Table 1. Comparison of intrusion detection systems.**

| Citation | Algorithm | Dataset | Advantages | Disadvantage |
|---|---|---|---|---|
| [8] | Auto-Encoder,Deep Belief Network | NSLKDD, KDDCup and CIDDS-001 | reaching to better result in binary classification on KDDCup and CIDDS-001 datasets | Achieving only 76.64% for accuracy in binary classification and TPR for some attacks reach to 0 |
| [13] | Bidirectional Long Short Term Memor(BiDLSTM) | NSLKDD | Obtaining a higher accuracy, recall, and F-score than the conventional LSTM | Requiring more training time |
| [26] | CNN-BiLSTM | NSL-KDD and UNSW-NB15 | Using (OSS) to reduce the noise samples in majority category which lead to reduce training time | Lower detection rate for other types of attacks such as (Normal, Dos, Prob, R2L) |
| [31] | CNN-O- LSTM | DARPA1998, DARPA LLS DDoS-1.0, CICIDS2017, NSL-KDD and KDD cup | DDoS detection model through deep learning methods | Optimization for LSTM donot improve some metrices which obtains a very low specificity of 51% in all datasets |
| [11] | RNN | NSL-KDD | Reducing the number of features, which lead to reduce preprocessing time | They don't apply multiclass classification and use the same dataset for training and testing don't use testing dataset of NSLKDD (KDDTest+) |
| [41] | PCA-PNN | KDD99 | The computation of data is greatly reduced as features reduction from 122 to 6 which lead to reduce the detecting time | Using minimum instances for training and testing process so most dataset not covered in results |
| [42] | DSN | NSL-KDD | Combination the benefits of four machine learning techniques | Despite using oversampling give minimum detection rate for R2L and U2R attacks |
| [32] | WILS | CIDDS-001, UNSWNB15, KDD-cup99 | Optimization LSTM using whale algorithm help in gets significant results in in accuracy, precision, and recall | They Don't include performance in multi-class attacks |
| [25] | BLSTM-CNN | NSL-KDD dataset | Attention mechanism is used to obtain features which are more related for malicious traffic detection | Lower detection rate for U2R and R2L attacks |
| [34] | OCNN-HMLSTM | NSL-KDD, ISCX-IDS and UNSWNB15 | Author Implemented hierarchical Multi-scale LSTM (HMLSTM) for effective extraction and learning of spatial-temporal features which lead to achieve binary classification accuracy of 90% | False-positive rates reach 9.92% in all attack classes |
| [21] | LSTM | CIDDS-001 | LSTM achieved a reasonable accuracy of 0.85 in multi-class classification | Binary classification isn't implemented and there isn't any comparison with the traditional classifiers. |
| [22] | LSTM | KDD-cup99 | Applying principal component analysis give best accuracy in binary and multiclass classification | Five categories of attacks are grouped into three categories of attacks where we can not measure performance of another attacks. |

## 3.1 Chaotic map

Since the last decade, chaotic maps have been widely appreciated in the field of optimization due to their dynamic behaviour which helps optimization algorithms explore the search space more dynamically and globally [43]. Chaotic maps are ten mathematical functions that are used for the generation of chaotic sequences. In this paper, iterative map developed in [44] is used instead of random sequences. It has been tested before in [20] and gave better results than other chaotic maps, It is defined as follows:

$$x_{i+1} = sin(\frac{a*pi}{x_i}) \tag{1}$$

Where $a = \in (0, 1)$ and $pi = 3.14$.

## 3.2 Butterfly Optimization Algorithm (BOA)

BOA is a swarm optimization algorithm that was inspired from nature and mimics the foraging behaviour of social butterflies [17]. BOA searches both locally and globally for the best solution for a given problem. In BOA, information is propagated to all other search agents (solutions) using fragrance to form a collaborative social network. All previous skills in BOA will help in optimization and searching for optimal parameters. In nature, butterflies use sensors to sense or smell fragrance. Each butterfly scatters a different amount of fragrance according to its fitness. A butterfly emits a strong fragrance with intensity when it moves. An algorithm for standard BOA is shown in Algorithm 1. The fragrance of each butterfly can be defined as follows.

$$pf_i = cI^a \tag{2}$$

Where $pf_i$ represents the perceived magnitude of fragrance, $I$ is fragrance intensity. The parameters $a$ and $c$ are the power exponent and the sensor modality, respectively.

The parameter ($a$) is the power exponent defining the variation of fragrance absorption, which affects the butterfly's ability to find the best solution. If $a=1$, this indicates no absorption of fragrance. That is, the other butterflies will sense all amounts of the fragrance emitted by a particle butterfly. If $a=0$, then the fragrance emitted by a particle butterfly is not perceivable to any other butterflies. We can see the role of ($a$) in optimization, so we use the following equation developed in [28] to balance the BOA search capabilities.

$$a(t) = a_s - (a_s - a_f) \times sin((\frac{\pi}{\mu}) \times (\frac{t}{T_{max}})^2) \tag{3}$$

Where $a_s$ and $a_f$ are the initial and final values of $a$, $\mu$ is the tuning parameter and $T_{max}$ is the maximum number of iterations. A value of sensor modality $c$ in the range [0, 1]. Its value can be updated in an iterative BOA process as follows,

$$c_{t+1} = c_t + (\frac{0.025}{c_t \times T_{max}}) \tag{4}$$

Where $T_{max}$ is the maximum number of iterations and initial value of $c$ is 0.01.

Each butterfly emits fragrance when it moves and the other butterflies are attracted to it according to its magnitude of fragrance. This process is called a global search and can be defined as follows

$$x_i^{t+1} = x_i^t + (r^2 \times g^* - x_i^t) \times f_i \tag{5}$$

Where $x_i^t$ is a vector which represents the butterfly (solution) at iteration $t$, $g^*$ is the overall best solution, $r$ is a random number in [0, 1] and $f_i$ is a fragrance of ith butterfly. When the butterflies fail to sense the fragrance of the other butterflies, they move randomly in the search space. The process is called local search and it can be defined as follows.

$$x_i^{t+1} = x_i^t + (r^2 \times x_j^t - x_k^t) \times f_i \tag{6}$$

Where $x_j^t, x_k^t$ are two vectors that represent two different butterflies in the same population.

**Algorithm 1** Butterfly optimization algorithm

```
1: Set the initial values of the population size n (butterflies),
parameters a (power exponent), c sensory modality, switch probability
ρ, and the maximum number of iterations Max_itr.
2: Set t := 0.        ▷ Counter initialization.
3: for (i = 1 : i ≤ n) do
```

```
4:    Generate an initial population (butterflies) x⃗ᵢᵗ randomly.
5:    Evaluate the fitness function of each butterfly (solution) f(x⃗ᵢᵗ).
6:    Calculate the fragrance for x⃗ᵢᵗ as shown in Eq 2.
7:    Assign the overall best butterfly (solution) g⃗*.
8: end for
9: repeat
10:   Set t = t + 1.
11:   for (i = 1 : i ≤ n) do
12:     Generate random number r, r ∈ [0, 1].
13:     if (r < ρ) then
14:       Move butterflies towards the best butterfly g⃗* as shown
              in Eq 5.           ▷ Global search.
15:     else
16:       Move butterflies randomly as shown in Eq 6.     ▷ Local search.
17:     end if
18:     Evaluate the fitness function of each butterfly (solution) f(x⃗ᵢᵗ).
19:     Assign the overall best solution g⃗*.
20:   end for
21:   Update the value of parameters a, c.
22: until (t > Maxᵢₜᵣ).     ▷ Termination criteria satisfied.
23: Produce the best solution g⃗*.
```

## 3.3 Chaotic butterfly optimization algorithm (CBOA)

CBOA is a modified version of BOA that uses chaotic maps instead of random variables in Eqs 6 and 7 to update butterfly positions. Thus enhancing BOA's accuracy, as described in [20]. For global search, Eq 7 can be changed as follows.

$$x_i^{t+1} = x_i^t + (C^2 \times g^* - x_i^t) \times f_i \tag{7}$$

Where $x_i^t$ is a vector which represent the butterfly (solution) at iteration $t$, $g^*$ is the overall best solution, $C$ is a chaotic number and $f_i$ is a fragrance of ith butterfly. For local search, Eq 6 can be updated as follows.

$$x_i^{t+1} = x_i^t + (C^2 \times x_j^t - x_k^t) \times f_i \tag{8}$$

Where $x_j^t, x_k^t$ are two vectors that represent two different butterflies in the same population.

## 3.4 Particle swarm optimization (PSO)

Kennedy and Eberhart proposed PSO as one of the bio-inspired algorithms in 1995 [45]. PSO is established by certain species' social foraging behaviour, such as schooling behaviour in fish and flocking behaviour in birds. An algorithm for standard PSO is shown in Algorithm 2. PSO consists of particles, each of which has its own velocity and position. In PSO, each particle moves to the best local position *Pbest* and the best global position *gbest*, where *Pbest* is the particle's best local location and *gbest* is the best position from all the best local positions. Each particle has a velocity defined as follows.

$$v_i^{t+1} = W \times v_i^t + c_1 \times r_1 \times (pbest_i^t - x_i^t) + c_2 \times r_2 \times (gbest_i^t - x_i^t) \tag{9}$$

Where i = 1;2….S; and S is swarm size,$c_1$ and $c_2$ are factors of constant cognitive and social scaling. W is inertia weight was added to boost performance [28]. W is calculated by the

following equation.

$$W(t) = W^{max} - \frac{(W^{max} - W^{min}) \times T_i}{T_{max}} \tag{10}$$

Where $T_{max}$ is the maximum number of iterations, $T_i$ is a current iteration. $W^{max}$ and $W^{min}$ is the maximum and minimum value of inertia weight respectively.

The location of the particle at iteration $t$ is calculated as follows.

$$x_i^{t+1} = x_i^t + v_i^{t+1} \tag{11}$$

**Algorithm 2** Particle swarm optimization

```
1: Input: Randomly initialized position and velocity of Particles: x_i^t
and v_i^t
2: Output: Position of the approximate global minimum X*
3: while terminating condition is not reached do
4:   for i = 1 to number of particles do
5:     Calculate the fitness function f
6:     Update personal best and global best of each particle
7:     Update velocity of the particle using Eq 9
8:     Update the position of the particle using Eq 11
9:   end for
10: end while
```

### 3.5 Long short term memory (LSTM)

LSTM is an extension of RNN that able to learn long-term dependencies. The LSTM architecture is more complicated than the RNN architecture; it has four hidden layers that use gates to add and remove cell state information [46].

For one LSTM cell, at time step t, the forget, input and output gates are represented by $i_t$, $O_t$, $f_t$, respectively, as shown in Fig 1 which discussed before in [47]. Forget gate decides which information will be deleted from the cell state based on $h_{t-1}$ and $x_t$. The input gate determines which information from the current state will be stored in the cell state and updates it using the 'tanh layer' to generate a vector of new contender values. The final output gate decides how

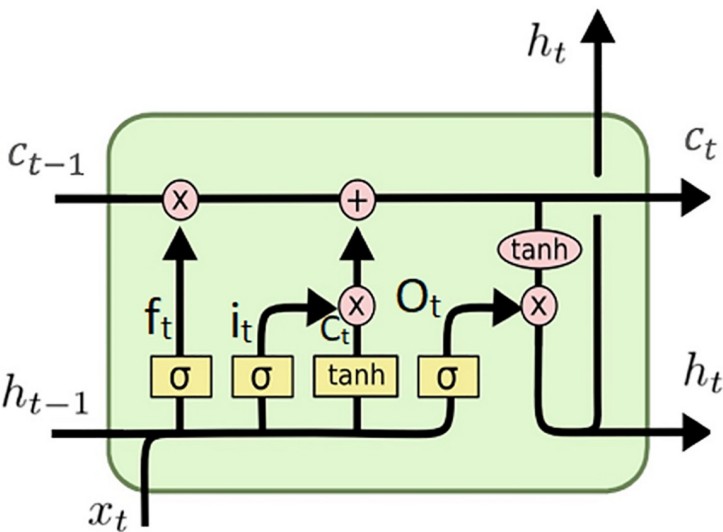

**Fig 1. The architecture of LSTM cell.**

the output should look and passes it through the 'tanh layer' to the next neuron. The following equations mathematically describe the relationship between the inputs and outputs at time $t$ and $t - 1$:

$$f_t = \sigma(W_f \cdot [h_{t-1}, x_t] + b_f). \tag{12}$$

$$i_t = \sigma(W_i \cdot [h_{t-1}, x_t] + b_i) \tag{13}$$

$$o_t = \sigma(W_o \cdot [h_{t-1, x_t}] + b_o) \tag{14}$$

$$g_t = tanh(W_c \cdot [h_{t-1}, x_t] + b_c) \tag{15}$$

$$C_t = f_t * C_{t-1} + i_t * g_t \tag{16}$$

$$h_t = o_t * tanh(C_t) \tag{17}$$

Where $C$ denotes the cell state The activation functions are defined by sigma (the sigmoid function) and tanh. $x$ is the input vector, and $h_t$ is the output vector. The weights and biases parameters are represented by $W$ and $b$, respectively. A tanh layer generates a vector of new candidate values, $g$, which can be added to the state.

In this paper, we develop a deeper LSTM network with four hidden layers and two input and output layers. It starts by mapping inputs to their representations using the feature input layer. It then feeds the sequence to two double LSTM layers. LSTM outputs are then fed to two fully connected layers with the rectified linear unit (RELU) as an activation function. Finally, the fully connected layers learn and compile the extracted data from the LSTM layer to form a final output that passes through an output layer for classification. Fig 2 displays a summary of the LSTM network architecture with four hidden layers as a first phase in the proposed algorithm.

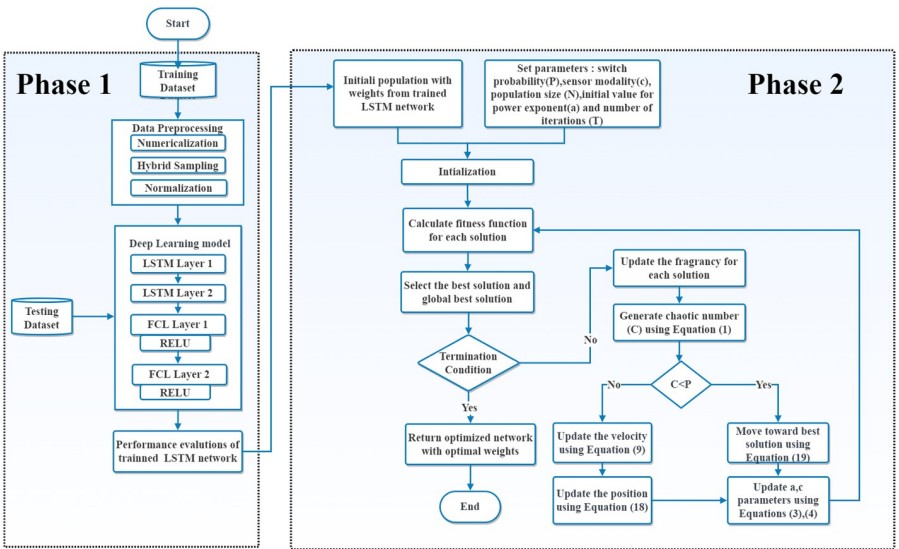

**Fig 2. The architecture of ILSTM.**

## 4 The proposed ILSTM algorithm

The proposed algorithm (ILSTM) consists of a hybrid LSTM network described in section 3.5 and the hybrid swarm algorithm CBOA and PSO, as briefed in Sections 3.3 and 3.4, respectively. The hybrid CBOA+PSO was used for optimising weights of the LSTM network, which helps in improving the training of the LSTM network in a minimum number of iterations. In general, the proposed ILSTM consists of two main phases as described in Fig 2.In the first phase, the LSTM is traditionally trained to get the best parameters and weights of its internal network architecture. In second phase, the hybrid CBOA+PSO (see Algorithm (3)) was used for optimising weights of the trained LSTM network to further find the optimal weights which can improve the accuracy in both binary and multi-class classification1ion while taking a fewer iteration. More details are in the following sub-sections and in the ILSTM algorithm given in Algorithm (3).

### 4.1 Phase 1: Training LSTM network

In order to obtain better weights from the trained network than random weights for phase initialization, we first implemented a deeper LSTM network. The LSTM network was trained with four hidden layers: (LSTM layer 1 + LSTM layer 2) followed by two fully connected Layers (FCL) with rectified linear unit (Relu). The parameters of LSTM network are described in Table 4. When the training accuracy of the LSTM network did not show improvement, the proposed algorithm uses phase 2 to improve performance in a fewer number of iterations.

### 4.2 Phase 2: LSTM network optimization and acceleration

In this phase, by integrating the capabilities of the individual CBOA and PSO algorithms, we were able to combine their benefits for accurately optimizing the weights of an LSTM network. In this case, PSO is employed for the local search for optimal weights while CBOA is used for the global search for optimal weights. The following steps explain how both algorithms were used to optimize the weights of the LSTM network.

1. **Generation of initial population** The proposed algorithm ILSTM initiates with weights obtained by the conventional LSTM in phase 1, and some parameters are used for CBOA, such as switch probability (P), sensor modality (c), and power exponent (a), and other parameters are used for PSO, such as minimum and maximum values of velocity inertia weight ($Wmin$, $Wmax$), and constant cognitive factors ($c1$, $c2$), as well as a number of iterations (T) and population size (N) from Table 5. At each iteration values of power exponent and sensor, modality are updated based on the current iteration. CBOA and PSO are combined in all steps only in position updating, CBOA is used for global search and PSO is used for local search.

2. **Definition of fitness function** The fitness function of the proposed algorithm is the maximization accuracy of ILSTM which is calculated using the ACC equation in 20.

3. **Updating weights of network** At each iteration, ILSTM updates LSTM network with new weights and the fragrance of each solution is calculated.

4. **Position updating** Each solution in the population moves to next position according to the value of the chaotic number generated by Eq 1. If value of $c$ is greater than $P$, ILSTM uses the following equation for updating the position in local search.

$$x_i^{t+1} = x_i^t + v_i^{t+1} \tag{18}$$

Where $v_i^{t+1}$ is velocity defined before in Eq 9. If value of $c$ is less than $P$, ILSTM utilises the

following equation for updating position in global search.

$$x_i^{t+1} = x_i^t + (C^2 \times g^* - x_i^t) \times f_i \tag{19}$$

At each iteration, ILSTM selects optimal solutions(weights) according to a maximum value of the fitness function (the maximum value of accuracy).

5. **Termination condition** When ILSTM algorithm reaches to the maximum number of iterations, optimal weights with the best fitness function are produced. Finally, an optimized ILSTM network with optimal weights was generated.

**Algorithm 3** Proposed algorithm (ILSTM)

```
1: Set the initial values of the population size S (butterflies),
parameters a (power exponent), c sensory modality, switch probability
ρ, and the maximum number of iterations Max_itr.
2: Get an initial population (weights) from trained LSTM network x_i^t.
3: Set t := 0.       ▷ Counter initialization.
4: for (i = 1 : i ≤ S) do
5:    Evaluate the fitness function of each butterfly (weight) f(x_i^t).
6:    Calculate the fragrance for x_i^t as shown in Eq 2.
7:    Assign the overall best butterfly (weight) g*.
8: end for
9: repeat
10:   Set t = t + 1.
11:   for (i = 1 : i ≤ S) do
12:     Generate chaotic number C by Eq 1
13:     if (C < ρ) then
14:       Move butterflies towards the best butterfly g* as shown
          in Eq 19.       ▷ Global search.
15:     else
16:       Update the velocity using Eq 9.
17:       Update the position by Eq 18.       ▷ Local search.
18:     else if
19:     Evaluate the fitness function of each butterfly (weight) f(x_i^t).
20:     Assign the overall best weights g*.
21:   end for
22:   Update the value of a according Eq 3.
23:   Update the value of c according Eq 4.
24:   Update the value of W according Eq 10.
25: until (t > Max_itr).       ▷ Termination criteria satisfied.
26: Produce the best solution (optimal weights) g*.
27: Produce optimized LSTM network (ILSTM).
```

## 5 Experimental setup

This section gives details about the experimental setup under which the experimental evaluations, in the next section, are conducted. Firstly, all experiments have been conducted on a laptop with an Intel(R) Core(TM) i5-6300U CPU@ 2.50 GHz and 8.00 GB of RAM and the proposed algorithms were implemented using Matlab R2020a running on Windows 10.

In Section (5.1), an overview of the performance metrics used to assess the quality of the proposed algorithm is given. The section then gives a description and the preprocessing of the two public datasets (NSL-KDD 2009, LITNET-2020) used for the evaluation process. Finally, we test our proposed algorithm on a modern dataset, LITNET-2020, to ensure its efficiency. On the other hand, Table 4 displays a summary of the LSTM network architecture. We compare the algorithm's performance with state-of-the-art and deep learning methods trained and tested on the same dataset (i.e., the NSL-KDD dataset).

## 5.1 Performance metrics

Nine performance metrics, accuracy (ACC), detection rate (DR), false alarm rate (FAR), precision (Prec), specificity (SPC), f-measure, false negative rate (FNR), mathematic correlation coefficient (MCC), and kappa coefficient, were selected to evaluate the performance of ILSTM [34]. A mathematical representation of all measures can be calculated based on four performance measurements, true positive (TP), false positive (FP), true negative (TN), and false negative (FN). These four measures were collected from the confusion matrix [48].

1. **Accuracy**: the percentage of correctly classified instances to the total number of instances, defined as follows.

$$ACC = \frac{TP + TN}{TP + TN + FN + FP} \tag{20}$$

2. **Recall(DR)**: the equivalent TPR. It is the percentage of instances identified correctly over the total number of anomaly instances, it can be derived as follows.

$$DR = \frac{TP}{TP + FN} \tag{21}$$

3. **SPC**: is computed as follows.

$$Specifity = \frac{TN}{TN + FP} \tag{22}$$

4. **Prec**: is calculated as follows.

$$Precision = \frac{TP}{TP + FP} \tag{23}$$

5. **F-measure**: computed as follows.

$$F - measure = 2*\frac{Precision*Recall}{Precision + Recall} \tag{24}$$

6. **FAR**: known as FPR, the percentage of the number of normal instances which are misclassified as anomalies is divided by the total number of normal instances, can be computed as follows.

$$FAR = \frac{FP}{FP + TN} \tag{25}$$

7. **FNR**: can be computed as follows.

$$FPR = \frac{FN}{FN + TP} \tag{26}$$

8. **MCC**: varies between −1 and 1 where the best binary classifier obtains positive 1 and worst classifier obtains negative 1. It is computed as follows

$$MCC = \frac{(TP*TN) - (FP*FN)}{\sqrt{(TP + FP)(TP + FN)(TN + FP)(TN + FN)}} \qquad (27)$$

9. **Kappa coefficient**: is used to check whether the classifier can process imbalanced data classes successfully. It is calculated as follows.

$$kappa = \frac{absolute - Expect}{1 - Expect} \qquad (28)$$

where Absolute = Accuracy and

$$Expect = \frac{A + B}{(TP + TN + FP + FN)} \qquad (29)$$

values of A and B can be obtained as

$$A = \frac{(TP + FN)(TP + FP)}{(TP + TN + FP + FN)} \qquad (30)$$

$$B = \frac{(FP + TN)(FN + TN)}{(TP + TN + FP + FN)} \qquad (31)$$

## 5.2 Dataset 1: NSL-KDD dataset

The NSL-KDD dataset is a refined version of the KDD cup [49]. It has a fair distribution of all types of attacks [50]. Many researchers employ the NSL-KDD to develop an effective intrusion detection algorithm, such as in [26, 34, 50]. The NSL-KDD includes 41 attributes that are classified as normal or attack traffic [49]. The NSL-KDD is divided into a training dataset (KDDTrain+) and two testing datasets, KDDTest+ and KDDTest-21. All of these datasets have normal records and four types of attack records, such as probe, remote to local (R2L), denial of service (Dos), and user to root (U2R). In this paper, all of the KDDTrain+ dataset is used for training, and all of two other datasets (i.e., KDDTest+ and KDDTest-21) are used for testing, where the training dataset represents 80% of the NSL KDD dataset and the testing dataset represents 20% of the NSL KDD dataset as shown in Table 2.

**5.2.1 Dataset preprocessing.** KDDTrain+, KDDTest+, and KDDTest-21 datasets are preprocessed before being used for training and testing the LSTM network and the proposed ILSTM. We apply preprocessing step on raw dataset to better make full use of domain knowledge of network traffic. It contains three processes: (1) mapping symbolic features to numeric

**Table 2. NSL-KDD dataset description.**

| Datasets | Normal | Dos | Probe | R2L | U2R | Total |
|---|---|---|---|---|---|---|
| KDDTrain+ | 67343 | 45927 | 11656 | 995 | 52 | 125973 |
| KDDTest+ | 9711 | 7458 | 2421 | 2754 | 200 | 22544 |
| KDDTest-21 | 2152 | 4342 | 2402 | 2754 | 200 | 11850 |

**Table 3. Transformation of symbolic features in NSL-KDD.**

| Symbolic features | transform number |
|---|---|
| protocol_type | tcp = 1,udp = 2,icmp = 3 |
| service | auth = 1, bgp = 2, courier = 3, cenet_ns = 4, ctf = 5, daytime = 6, discard = 7, domain = 8, domain_ u = 9, echo = 10, eco_ i =11,ecr_ i = 12, efs = 13, exec = 14, finger = 15, ftp = 16, ftp_data = 17,gopher = 18, hostname = 19, http = 20, http_443 = 21, http_8001 = 22, imap4 = 23, IRC = 24, iso_tsap = 25, klogin = 26, kshell = 27, ldap = 28, link = 29, login = 30,mtp = 31, name = 32, netbios_dgm = 33, netbios_ns = 34, netbios_ssn = 35, netstat = 36, nnsp = 37, nntp = 38, ntp_ u = 39, other = 40, m_dump = 41, pop_2 = 42, ppop_3 = 43, printer = 44, private = 45, red_ i = 46, remote_job = 47, rje = 48, sshell = 49, mtp = 50, sql_net = 51, ssh = 52, sunrpc = 53, supdup = 54, systat = 55, telnet = 56, tftp_ u = 57, tim_ i = 58, time = 59, urh_ i = 60, urp_ i = 61, uucp = 62, uucp_path = 63, vmnet = 64, whois = 65, X11 = 66, Z39_50 = 67 |
| flag | SF = 1, S0 = 2, REJ = 3, RSTR = 4, SH = 5, RSTO = 6, S1 = 7, RSTOS0 = 8, S3 = 9, S2 = 10, OTH = 11 |

values; (2) sampling imbalanced classes in the dataset; and (3) normalizing features that have a large scale.

1. **Data transformation** The NSL-KDD dataset has 38 numeric features and 3 non-numeric features such as "protocol-type," "service," and "flag". As LSTM classifier accepts only numeric values, we first convert non-numeric features, as in [51, 52], where we replace every single value with an integer in order to handle non-numeric features as in Table 3. One-hot encoding makes our training data more useful and expressive, and it can be rescaled easily. By using numeric values, we can more easily determine the probability of our values. In particular, one hot encoding is used for our output values since it provides more nuanced predictions than single labels. Each value is converted to binary code, so a protocol type with three values (tcp, udp, and icmp) becomes 1, 2, and 3, which are recognised as [1, 0, 0], [0, 1, 0], and [0, 0, 1]. Finally, NSLKDD has 122 features.

   In order to further analyse NSL-KDD, we used SHAP analysis for indication of structural predictors (inputs) that have the strongest influence on the particular output. This was done by evaluating the effect of each feature on the target variable and indicating the importance of each feature in determining the final predicted outputs. The outcome of the SHAP analysis is given in Figs 3 and 4.

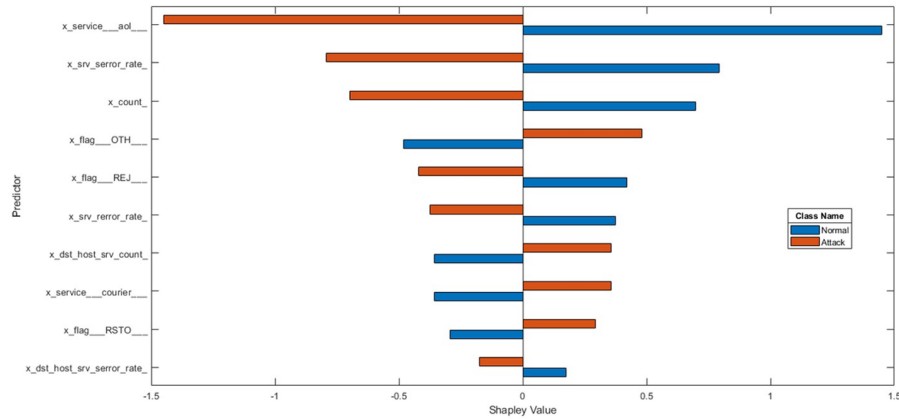

**Fig 3. Shap analysis for NSL-KDD in binary classification.**

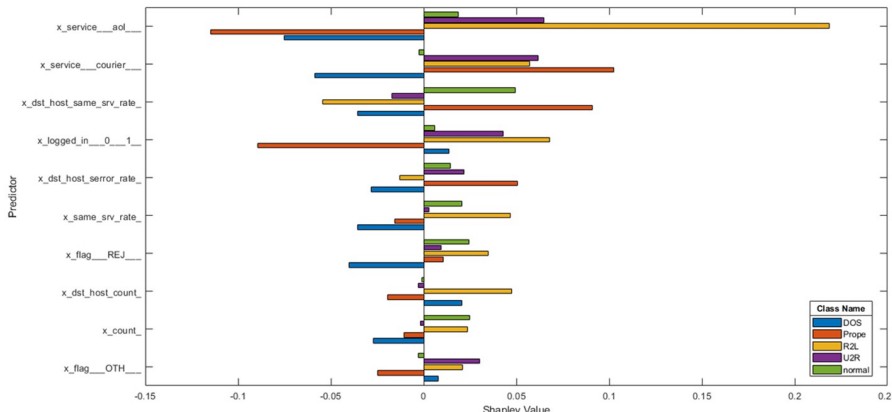

**Fig 4. Shap analysis for NSL-KDD in multi-class classification.**

2. **Dataset balancing using hybrid sampling** There are several aspects that might influence the performance of learning systems. One of these aspects is related to class imbalance, which occurs when training data have a larger number of examples for one class than other classes, such as in the NSL-KDD dataset. The classes in NSL-KDD network traffic data are not represented equally, where Normal and Dos have larger examples than other types of attacks such as U2R, R2L and Prob so these imbalanced data make a problem with classification as the prediction of the majority class is increased while the detection of the minority class is very low. Prior to [26], hybrid sampling was used, and the results were better than those of a standard dataset. Synthetic minority over-sampling technique (SMOTE) is an over-sampling method [53]. SMOTE forms new minority class examples by matching several minority class examples that lie together. SMOTE can avoid overfitting and make minority class boundaries spread through majority class space. To balance majority classes, the random under sampling (RUS) [54] technique is used to reduce the number of examples of the majority classes in the training dataset.

3. **Normalization** Some features in the NSL-KDD dataset, such as "duration," "src-bytes", and "dst-bytes" have a large scope between the minimum and maximum values, which can degrade the classification performance [55]. So, we applied the minimum-maximum normalization method [53] which maps features into the normalized range [0, 1]. This method can be defined as in Eq [32].

$$X_{norm} = \frac{X - X_{min}}{X_{max} - X_{min}} \tag{32}$$

Where $X_{min}$ and $X_{max}$ are the minimum and the maximum values of feature $x$.

**5.2.2 Parameter setting.** To determine the value of the parameters of the selected algorithms, we study the performance of LSTM network on NSL-KDD. Then the hybrid algorithms (i.e., CBOA+PSO) was used to optimize the weights of LSTM network and finally we evaluate the performance of the proposed algorithm ILSTM in binary classification (normal, anomaly) and five category classification (multi-classification) such as (Dos, Prope, R2L, and U2R). KDDTest+ dataset is used to determine the optimal parameters and network topology of the algorithm. These parameters and network topology are then applied to the KDDTest-21

**Table 4. Parameter setting for LSTM.**

| Parameter | Binary | Multi-class |
| --- | --- | --- |
| Optimizer | Adam | Adam |
| Learning rate | 0.001 | 0.01 |
| Hidden nodes for LSTM1 | 64 | 64 |
| Hidden nodes for LSTM2 | 32 | 32 |
| Hidden nodes for FCL1 | 8 | 16 |
| Hidden nodes for FCL2 | 4 | 8 |
| Epochs | 100 | 100 |
| L2Regularization | 0.01 | 0.0001 |
| Loss function | cross entropy | cross entropy |
| output layer activation | softmax | softmax |

and LTNET-2020 datasets. The parameters used in the simulation of the LSTM network are shown in Table 4, where:

1. The adaptive moment estimation (Adam) algorithm is used to update LSTM network's parameters. For the binary classification, the loss function was cross-entropy while for multi-classification the categorical cross-entropy was used. We applied regularization in range [0.01, 0.001], which came down to adding a cost to the loss function for large weights to ensure that our network does not overfit the data.

2. When the learning rate of the network is too high, the loss function of networks will oscillate without convergence. If the learning rate is too low, the slow convergence rate will hinder the updating of networks. Therefore, choosing an appropriate learning rate is very important for network performance optimization. As in Fig 5 we studied the impact of a set of learning rates [0.1, 0.01, 0.001, 0.0001] in binary and multi-classification on the KDDTest+ dataset and selected the best learning rate that achieves high accuracy.

3. An essential component of choosing the overall neural network architecture is determining the number of neurons in the hidden layers. Applying too few neurons in the hidden layers will result in a problem called underfitting. When too many neurons are used in the hidden layers, a problem known as overfitting occurs and training time is increased. In this paper, we assumed that the number of hidden neurons should be between the size of the input layer and the output layer in a network model, so we applied Eq 33 as in [56] to get the best

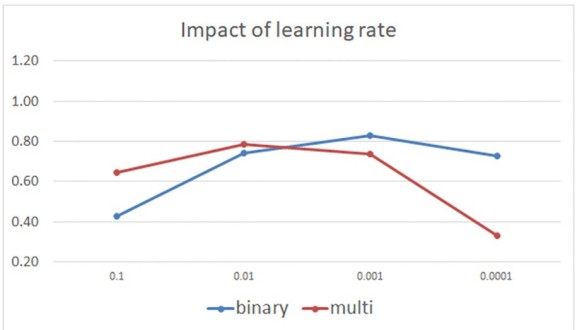

**Fig 5. Performance test on KDDTest+ with increasing learning rate.**

**Table 5. Parameter setting for CBOA and PSO.**

| Algorithm | Parameters | Definitions | Values |
|---|---|---|---|
| CBOA+PSO | $S$ | Search agents | 100 |
| | $Max_{it}$ | Maximum number of iteration | 40 |
| | $R$ | Independent runs | 10 |
| CBOA | $p$ | Switch probability | 0.5 |
| | $a$ | Power exponent | 0.1 |
| | $c$ | Initial value of Sensory modality | 0.01 |
| | $\mu$ | Tuning parameter | 2 |
| PSO | $Wmax$ | Min value of velocity inertia weight | 0.9 |
| | $Wmin$ | Max value of velocity inertia weigh | 0.2 |
| | $c_1 = c_2$ | Factors of constant cognitive | 2 |

values.

$$N_h = N_s(\alpha \times N_t(N_i + N_o)) \tag{33}$$

Where $N_i$ = number of input neurons, $N_o$ = number of output neurons, $N_s$ = number of samples in training data set, $\alpha$ = an arbitrary scaling factor usually be in the range [2, 10] and $N_t$ = the number order for hidden layer.

Table 5 studies parameters for the CBOA and PSO algorithms, which are applied in various research papers like as [17, 28].

## 5.3 Dataset 2: LITNET-2020 dataset

LITNET-2020 dataset is a relatively new dataset collected by LITNET (Lithuanian research and education network) academic network in Lithuania's real-time network traffic. It is a real-world and up-to-date flow-based network dataset [57] which is developed to test IDS systems. In this dataset, there were 85 network flow features and 12 attack types, a summary of the attacks and their instances are given in Table 6.

**Table 6. Summary of LITNET-2020 dataset.**

| Class | Size |
|---|---|
| Benign | 36,423,860 |
| SYN flood | 3,725,838 |
| Code red | 1,255,702 |
| UDP flood | 93,583 |
| Smurf | 59,479 |
| LAND DoS | 52,417 |
| W32.Blaster | 24,291 |
| HTTP flood | 22,959 |
| ICMP flood | 11,628 |
| Port scan | 6232 |
| Reaper worm | 1176 |
| Spam Botnet | 747 |
| Fragmentation | 477 |

**5.3.1 Dataset preprocessing.** By studying the LITNET-2020 dataset, it was found that it has many features, such as "fwd, opkt, and obyt," which have only one unique value. Additionally, it contains source and destination IP and port numbers which are distinct features and could not be used in attack detection. Therefore, there were only 16 features available for attack classification. Further pre-processing was done where all categorical features were encoded using label encoding. It was also noticed that some features, such as "sp" and "dp", have a large gap between the minimum and maximum values, which can degrade the classification performance. So, we applied the minimum-maximum normalization method 32 which maps features into the normalized range.

Further to that, we use SHAP analysis to explain the proposed algorithm's prediction by calculating the contribution of each feature to the prediction, because SHAP analysis shows the importance of each feature on the target variable [58]. The results of the SHAP analysis is illustrated in Fig 6.

**5.3.2 Dataset balancing using hybrid sampling.** LITNET-2020 dataset suffers from imbalance problem in class distribution, where the number of normal instances (benign) reaches 3/4 of the size of the dataset, as shown in Table 6. To address this problem, hybrid sampling, as given in point 2 in subsection 5.2.1, was applied to produce a balanced the datasets.

**5.3.3 Data splitting approach.** We divided the LITNET-2020 dataset to 60% for training process and 40% for testing and validation. We choose this approach after conducting a small experiment aiming to find out the best data-splitting approach. The results of this experiment are summarized in in Fig 7. As shown in this figure, we divided the LITNET-2020 dataset into 4 different training and testing sets. We then tested all of them and it was found that 60:40 set is the best approach.

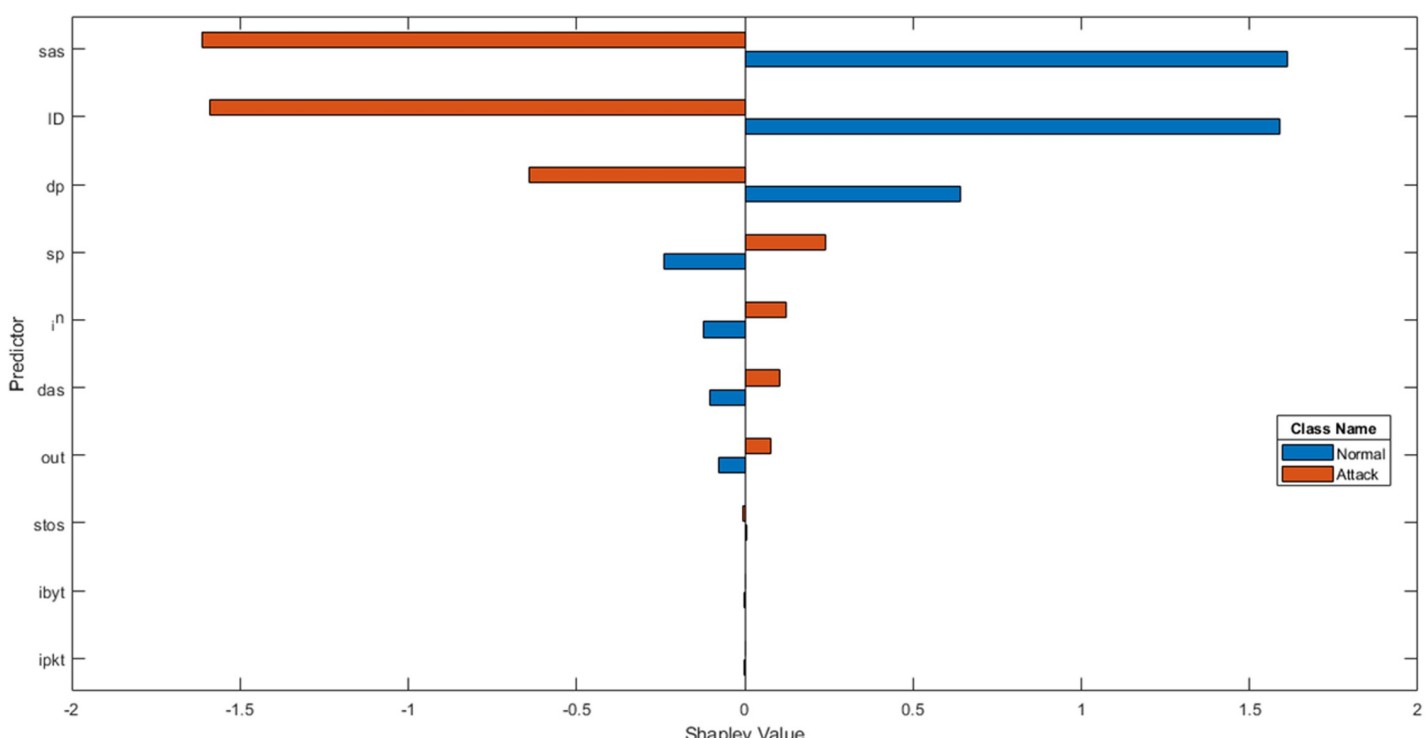

**Fig 6. SHAP analysis for LITNET-2020 dataset.**

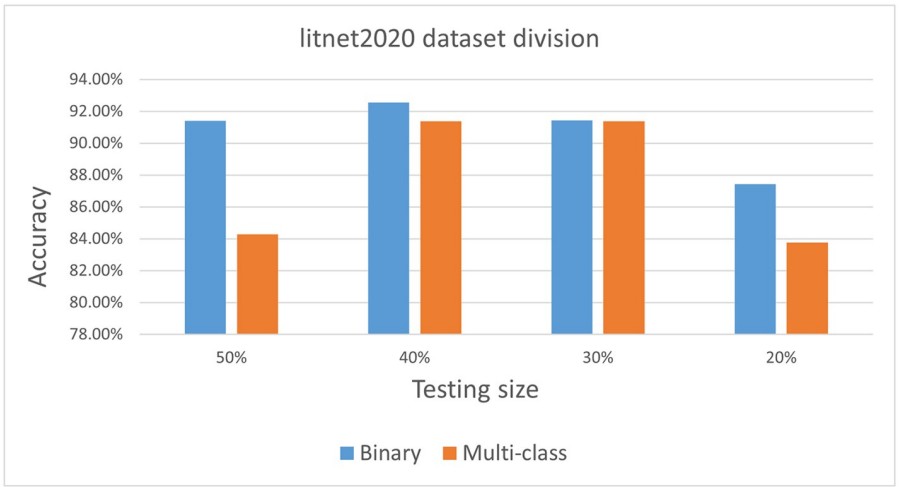

**Fig 7. LITNET-2020 data splitting approach.**

## 6 Results and discussion

The section reports the experimental results and their discussion which were conducted on the two datasets described above. For each dataset, two main experiments are implemented to study the performance of the proposed algorithm, ILSTM. In the first experiment, the proposed algorithm is investigated for binary classification (i.e., normal or malicious traffic), while in the second experiment, ILSTM is evaluated against multi-class classification (i.e., to differentiate among normal, dos, prob, U2R, or R2L). Also, in each experiment, (1) a statistical analysis (Wilcoxon test) was performed to show the significance of the ILSTM algorithm, and (2) a comparison with other deep learning and machine learning methods was conducted to demonstrate the efficiency of the ILSTM algorithm.

### 6.1 Experiment 1: ILSTM performance for binary classification on NSL-KDD dataset

The aim of this experiment is to assess the performance of the proposed ILSTM for intrusion detection in the case of classifying network traffic into normal or abnormal (i.e., binary classification). This was done on KDDTest+ and KDDTest-21 datasets, as detailed below.

**6.1.1 ILSTM Performance using KDDTest+ dataset.** To evaluate the performance of the proposed ILSTM, it is compared with the original LSTM and two optimized versions of LSTM using BOA and CBOA. A summary of the results of this experiment is given in Table 7. These results were recorded from an average of ten runs on the KDDTest+ dataset. It is clear from this table that the proposed ILSTM algorithm gave the best results achieving an accuracy of 91.31%, a specificity of 96.46%, and a FAR of 3.51% (which is a very important value for intrusion detection systems). Other best results are shown in bold text in this table. For detailed results of this experiment, the confusion matrix was reported in Fig 8.

Another experiment was conducted on the KDDTest+ dataset to investigate the relationship between the accuracy and the number of iterations of the proposed ILSTM and original LSTM. The results of this experiment were plotted in Fig 9. From this figure, it can be noticed that the ILSTM took iterations less than the LSTM but the latter achieved a higher accuracy. In this Fig, two curves are represented as follows: (a) a conventional LSTM achieved an accuracy

**Table 7. Comparison between LSTM, LSTM-BOA, LSTM-CBOA and ILSTM using KDDTest+ and KDDTest-21 in binary classification with average 10 runs.**

| Dataset | Method | ACC | DR | SPC | Preci | FAR | FNR | F1-Score | MCC | KAPPA |
|---|---|---|---|---|---|---|---|---|---|---|
| KDDTest+ | LSTM | 82.74 | **86.52** | 79.88 | 76.49 | 20.12 | **13.48** | 81.2 | 65.78 | 65.36 |
| | LSTM-BOA | 86.56 | 82.33 | 89.75 | 85.88 | 10.25 | 17.67 | 84.06 | 72.49 | 72.45 |
| | LSTM-CBOA | 88.62 | 79.85 | 95.25 | 92.85 | 4.75 | 20.15 | 85.81 | 77.05 | 76.40 |
| | ILSTM | **91.31** | 84.93 | **96.46** | **94.76** | **3.51** | 15.07 | **89.36** | **82.40** | **82.05** |
| KDDTest-21 | LSTM | 69.12 | **49.67** | 73.44 | 29.33 | 26.56 | **50.33** | 36.88 | 19.31 | 18.2 |
| | LSTM-BOA | 83.4 | 10.69 | 99.55 | 87.63 | 0.46 | 89.32 | 17.38 | 22.55 | 14.61 |
| | LSTM-CBOA | 84.59 | 18.13 | 99.34 | 87.22 | 0.66 | 81.87 | 28.16 | 32.23 | 24.17 |
| | ILSTM | **86.65** | 28.15 | **99.94** | **97.80** | **0.07** | 80.83 | **43.20** | **47.53** | **38.05** |

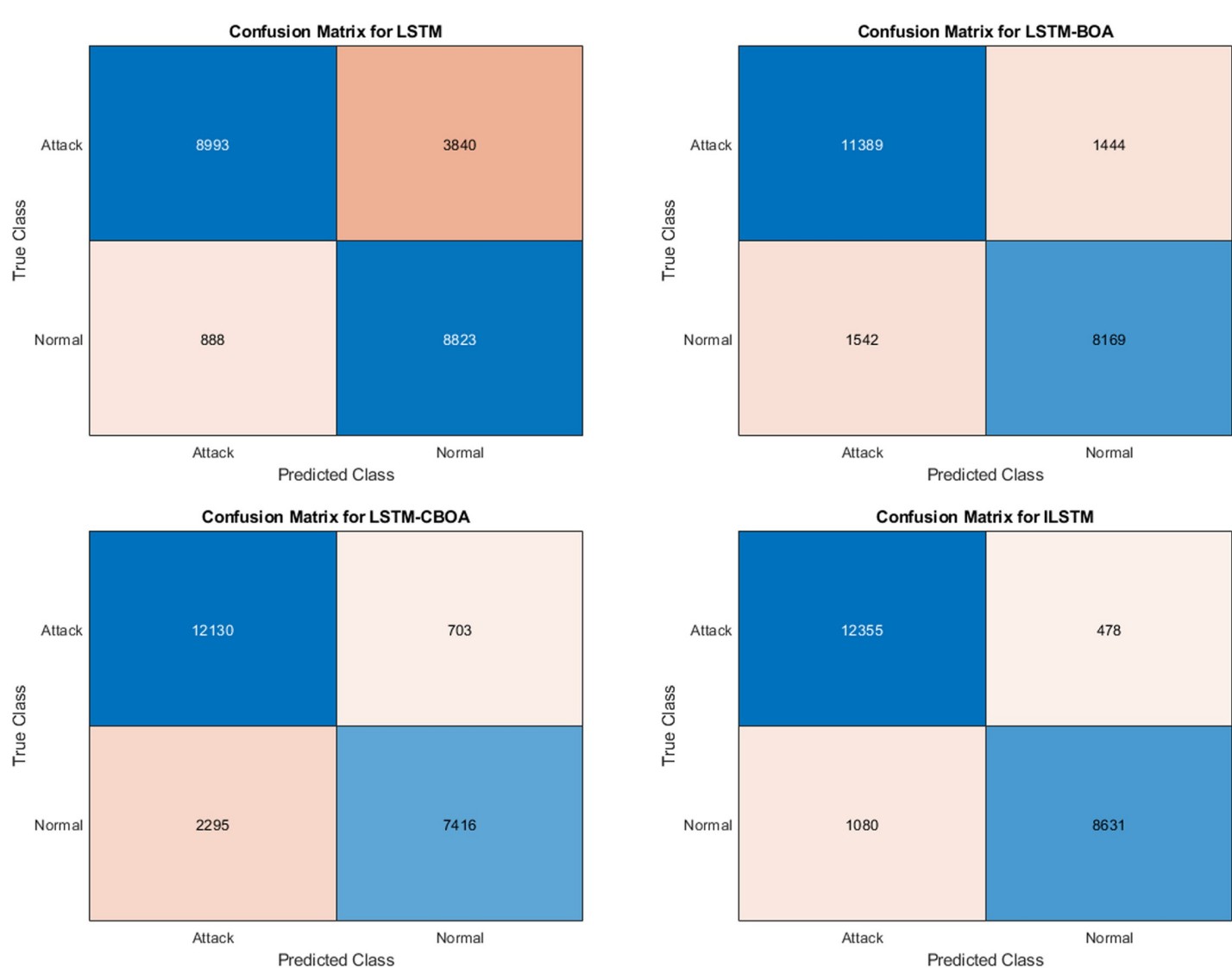

**Fig 8. Confusion matrices for KDDTest+ in binary classification.**

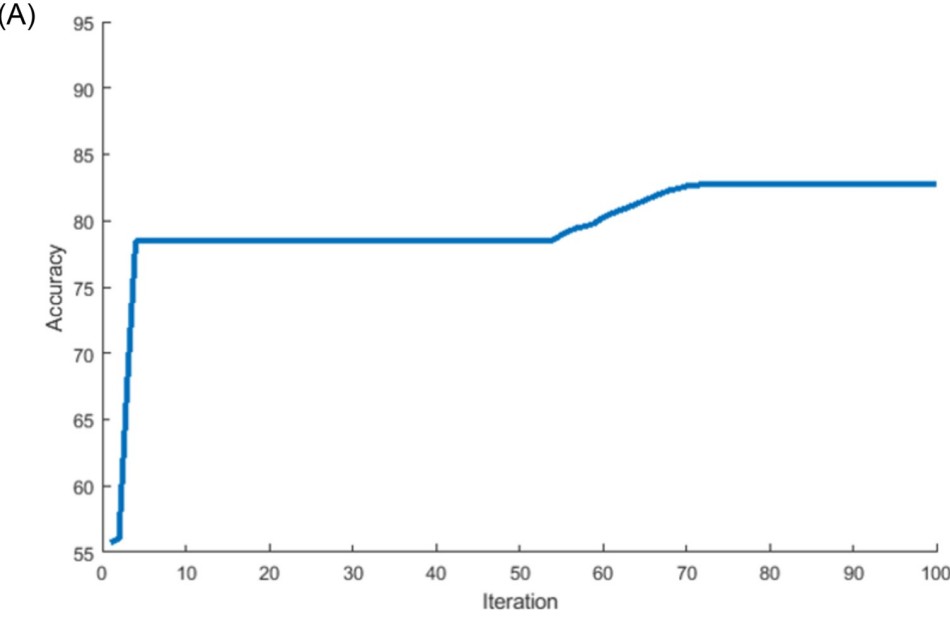

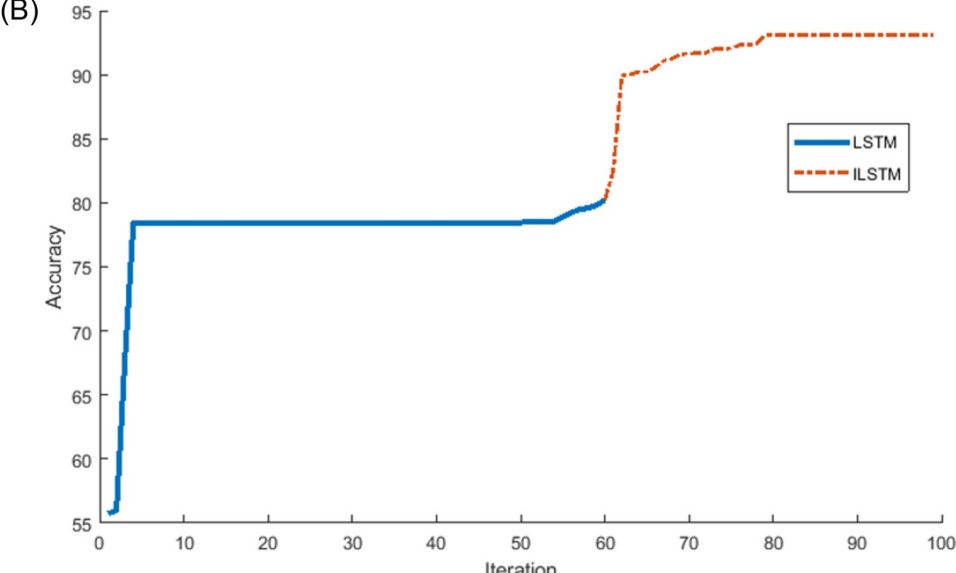

**Fig 9. Accuracy and number of iterations for LSTM and ILSTM using KDDTest+ in binary classifications.** A: LSTM. B: ILSTM.

of 82.74% at iteration 70 and did not improve between 70-100 iterations while Fig 9b shows that after using the CBOA and PSO for LSTM weights (i.e., ILSTM) the accuracy improved to 92.41% at iteration 80 and only increased by 1% from iteration 80-100.

**6.1.2 ILSTM Performance using KDDTest-21 dataset.** The same above experiment was conducted but using the KDDTest-21 dataset. The aim is to compare the proposed ILSTM with the original LSTM, LSTM-BOA, and LSTM-CBOA. A summary of the results is given in Table 7. An average of ten runs were used to get these results. Also, the confusion matrix for all

implemented algorithms in this experiment is shown in Fig 10. From these results, it could be concluded that the proposed ILSTM algorithm gave the best results in an accuracy of 86%, specificity 99%, precision 97.9%, and FAR 0.07 (which is a very important value for intrusion detection systems) and when it is small it means that the IDS is efficient. Other best results are shown in bold text in this table.

Using the KDDTest-21 dataset, we also investigated the relationship between the accuracy and the number of iterations of the proposed ILSTM and original LSTM. The results of this experiment were plotted in Fig 11. From this figure, it can be noticed that after applying the optimization phase using CBOA and PSO on LSTM, the accuracy improved with 18% from iterations 72-76, see Fig 11B while it remained constant at 68.95% for LSTM without any optimization see Fig 11A).

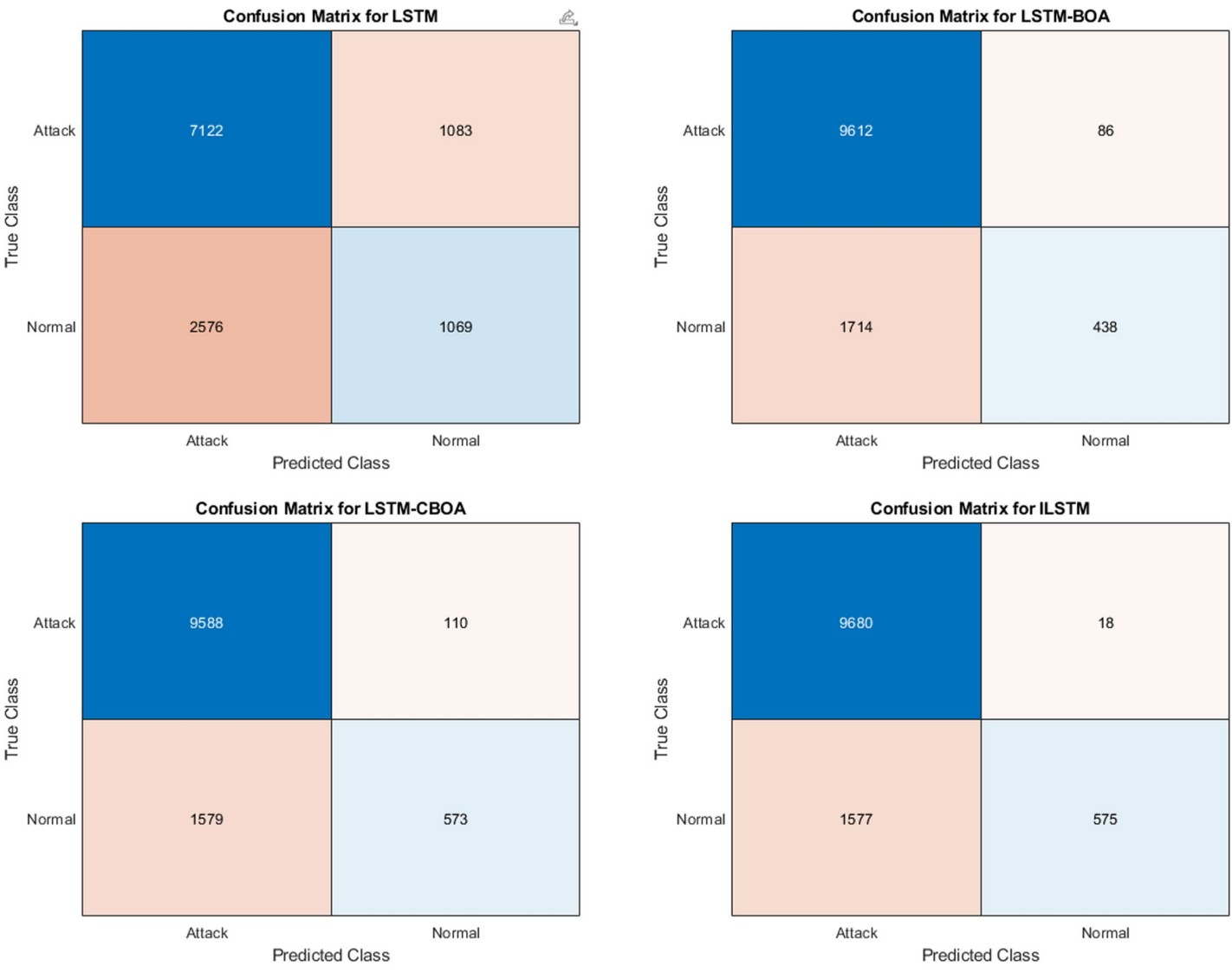

**Fig 10. Confusion matrices for KDDTest-21 in binary classification.**

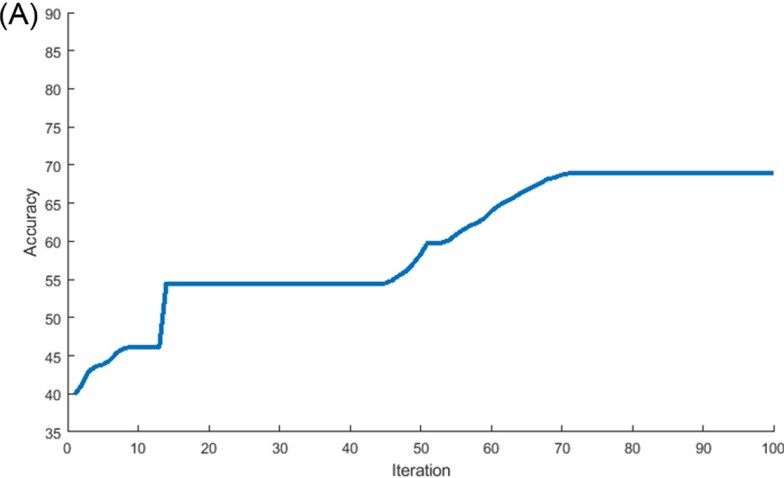

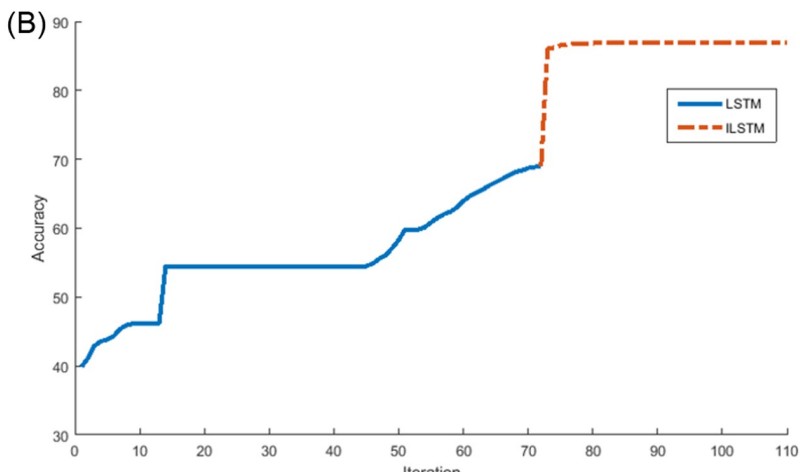

**Fig 11. LSTM vs ILSTM for KDDTest-21 for binary classification.** A: LSTM. B: ILSTM.

**6.1.3 Efficiency of ILSTM algorithm in binary classification.** Mean squared error (MSE) is used to calculate the difference between the actual value and the obtained value for measuring the performance in the optimization phase of LSTM. MSE is calculated using the following equation as in [59].

$$MSE = \frac{1}{n}\sum_{i=1}^{n}(x_i - y_i)^2 \tag{34}$$

Where $x_i$ is a vector of actual data, $y_i$ is a vector of predict data and $n$ is the number of instances in the testing dataset.

Fig 12 summarizes the results of the optimized weights of ILSTM. It can be seen that accuracy is not only improving but also the lowest value of MSE is reached in binary classification (normal or abnormal traffic) using two datasets: KDDTest+ in Fig 12(a) and KDDTest-21 in Fig 12(b).

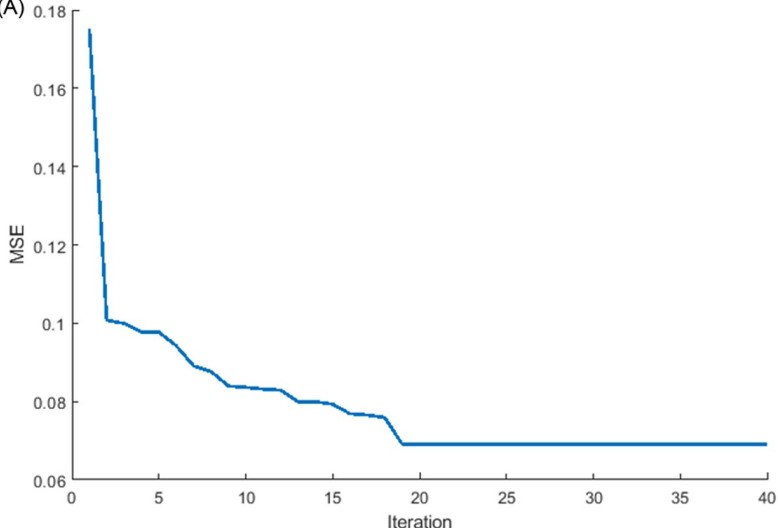

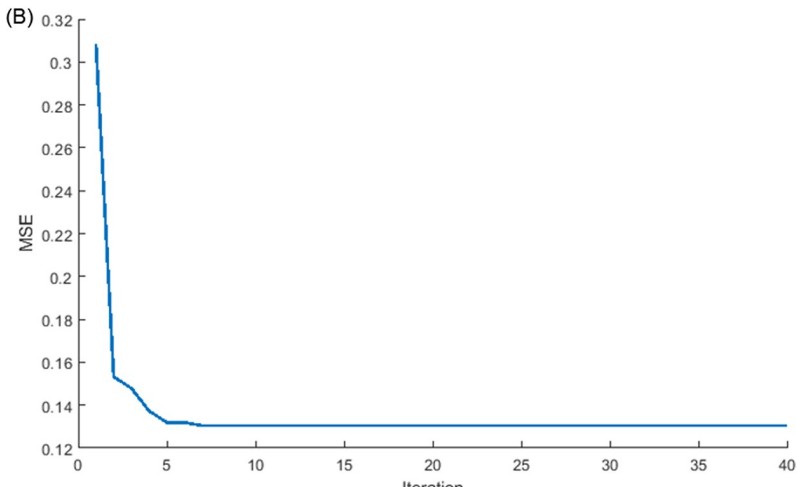

**Fig 12. MSE for ILSTM algorithm in binary classification.** A: KDDTest+. B: KDDTest-21.

**6.1.4 Wilcoxon signed-rank test for binary classification on KDDTest+ and KDDTest-21 datasets.** In this section, we implement the Wilcoxon signed-rank test to demonstrate the effectiveness of the proposed algorithm in binary classification using two testing datasets. As in Table 8, statistics results from Wilcoxon are defined between the accuracy of conventional LSTM and the accuracy of the proposed algorithm (ILSTM), where the mean difference is -10.2 for KDDTest + and equal to -22.86 for KDDTest -21. The value of z is 2.8031, and the p-value is 0.00512 for the KDDTest+ and KDDTest-21 datasets. The results in Table 8 show that the p = 0.00512 which is lower than the significance level (0.05), so the null hypothesis should be rejected and it means that there is a significant difference between the proposed ILSTM algorithm and the other algorithms. This statistical analysis confirms the numerical results reported above.

From Tables 7 and 8, three remarks can be noticed. Firstly, when using a chaotic map with BOA as in CBOA algorithm, the results of all evaluation metrics were improved. This

**Table 8. Statistic Test for KDDTest+ and KDDTest-21 in binary classification.**

| KDDTest+ | | | | | | KDDTest-21 | | | | | |
|---|---|---|---|---|---|---|---|---|---|---|---|
| LSTM | ILSTM | Sign | Abs | R | SignR | LSTM | ILSTM | Sign | Abs | R | SignR |
| 80.39 | 91.48 | -1 | 11.09 | 10 | -10 | 67.94 | 85.75 | -1 | 17.81 | 2 | -2 |
| 83.36 | 91.47 | -1 | 8.11 | 2 | -2 | 69.12 | 86.69 | -1 | 17.57 | 1 | -1 |
| 79.03 | 88.73 | -1 | 9.7 | 5 | -5 | 68.68 | 86.78 | -1 | 18.1 | 3 | -3 |
| 79.34 | 89.86 | -1 | 10.52 | 8 | -8 | 59.15 | 85.45 | -1 | 26.3 | 8 | -8 |
| 81.2 | 90.01 | -1 | 8.81 | 4 | -4 | 64.6 | 86.82 | -1 | 22.22 | 5 | -5 |
| 78.99 | 89.78 | -1 | 10.79 | 9 | -9 | 61.22 | 86.35 | -1 | 25.13 | 7 | -7 |
| 82.8 | 90.96 | -1 | 8.16 | 3 | -3 | 62.67 | 86.69 | -1 | 24.02 | 6 | -6 |
| 81.61 | 86.79 | -1 | 5.18 | 1 | -1 | 60.11 | 86.77 | -1 | 26.66 | 9 | -9 |
| 83.36 | 93.09 | -1 | 9.73 | 6 | -6 | 58.35 | 86 | -1 | 27.65 | 10 | -10 |
| 82.59 | 92.44 | -1 | 9.85 | 7 | -7 | 66.45 | 86.54 | -1 | 20.09 | 4 | -4 |

is due to the fact that a chaotic map increases the search space for new solutions while avoiding local minima. Secondly, when using Eq 11 of PSO algorithm in local search instead of Eq 8, the search for new solutions is improved because the velocity helps in searching for local and global best solutions. So, using CBOA and PSO in our proposed ILSTM algorithm improved the results of intrusion detection in both binary classification in most performance metrics such as (Acc, Spc, Prec, FAR, f1-score, MCC, and Kapp) on KDDTest+ and KDDTest-21 datasets. Finally, the optimization process improved the LSTM network with better results in performance metrics and statistical tests than the conventional LSTM.

**6.1.5 Comparison of the ILSTM algorithm with related methods.** In order to objectively evaluate the performance of ILSTM, we conducted a comparison with other deep and machine learning methods that were implemented in the intrusion detection literature. In this comparison, we used machine learning and deep learning methods reported in previous work such as [13, 14, 25]. The results of this comparison are reported in Fig 13 for machine learning methods and in Fig 14 for deep learning methods. From these figures, it can be concluded that our proposed algorithm outperformed all other algorithms in binary classification.

In addition, Table 9 shows the results of comparison with other methods suggested for binary classification. Those methods were used in [34]. From this table, it can be seen that the proposed ILSTM algorithm has achieved the best results in most measures, where the best values are shown in bold text.

## 6.2 Experiment 2: ILSTM performance for multi- class classification on NSL-KDD dataset

The aim of this experiment is to assess the performance of the proposed ILSTM for intrusion detection in the case of classifying network traffic into different types of attacks (i.e., multi-classification) where there are 5 classes of data Normal and 4 types of attacks (Dos, Prob, U2R, R2L). Three sub-experiments are conducted. The first and second are designed for the performance evaluation of ILSTM under the KDDTest+ and KDDTest-21 datasets, respectively while comparing it with the most related work. The third experiment was for comparison with the other related work under eight performance metrics.

**6.2.1 ILSTM performance using the KDDTest+ dataset.** This experiment aims to study the performance of ILSTM on accurately identifying four types of attacks (Dos, Prob, U2R,

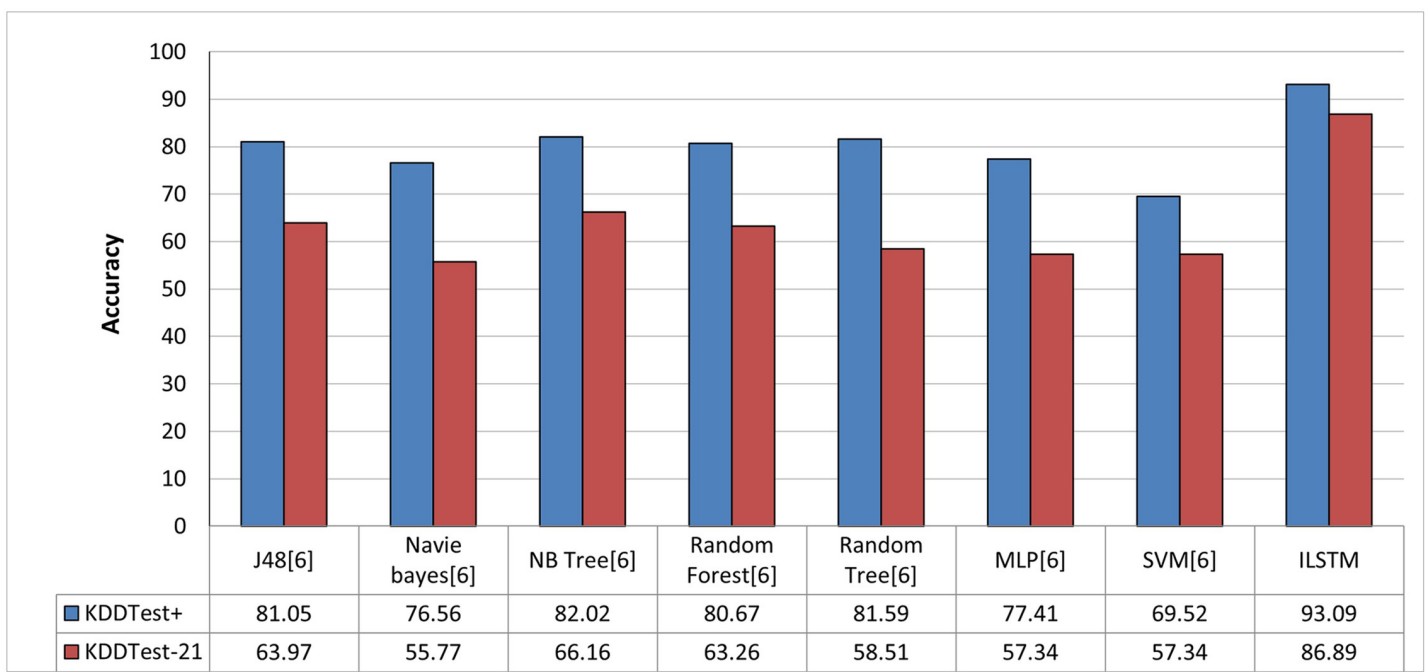

**Fig 13. Comparison between ILSTM and machine learning-based algorithms in binary classification.**

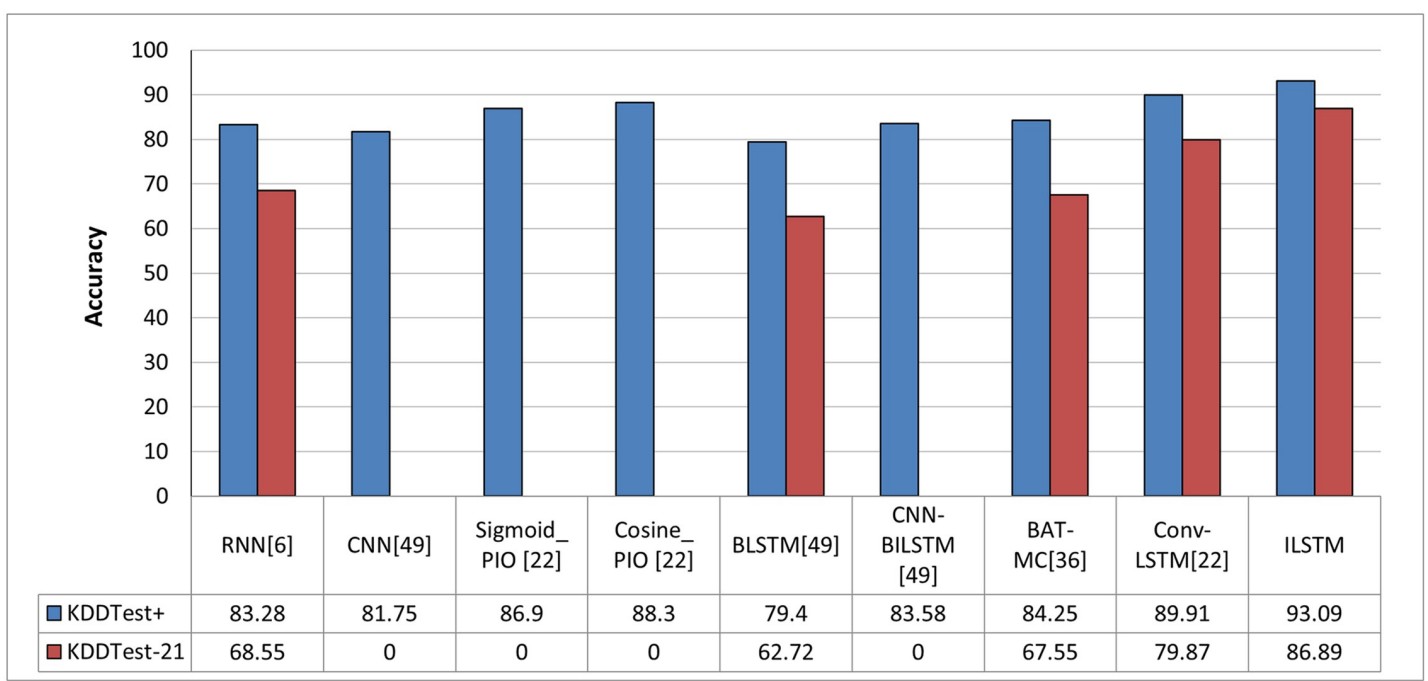

**Fig 14. Comparison between ILSTM and deep learning-based algorithms in binary classification.**

**Table 9. Comparison between other methods in literature using KDDTest+ for binary classification.**

| Method | ACC | Prec | Recall | F-Score | FAR | FNR | MCC | Kappa |
|---|---|---|---|---|---|---|---|---|
| NN [34] | 83.67 | 88.406 | 86.211 | 83.28 | 23.47 | 33.42 | 78.44 | 67.90 |
| DNN [34] | 89.33 | 86.67 | 92.56 | 90.67 | 12.67 | 11.2 | 81.01 | 75.67 |
| MSCNN [34] | 88.45 | 79.89 | 90.11 | 88.76 | 9.94 | 6.67 | 80.8 | 83.4 |
| ELM [34] | 81.33 | 81.94 | 84.91 | 81.95 | 26.02 | 56.28 | 69.79 | 77.47 |
| Conv-LSTM [34] | 89.94 | 80.501 | 88.87 | 88.77 | 11.8 | 9.95 | 78.9 | 83.34 |
| MSCNN [34] | 88.45 | 79.89 | 90.11 | 88.76 | 9.94 | 6.67 | 80.8 | 83.4 |
| OCNN [34] | 88.67 | 84.34 | 90.12 | 89.78 | 11.89 | 7.89 | 76.78 | 81.12 |
| HMLSTM [34] | 87.11 | 78.89 | 93.67 | 8.4 | 12.2 | 6.66 | 81.10 | 80 |
| OCNN-HMLSTM [34] | 90.67 | 86.71 | **95.19** | 91.46 | 8.86 | **5.78** | 82.22 | **86.33** |
| CNN-BILSTM [26] | 83.58 | 85.92 | 84.49 | 85.14 | NA | NA | NA | NA |
| ILSTM | **93.09** | **95.86** | 88.88 | **91.72** | **2.68** | 11.12 | **85.9** | **85.8** |

R2L) using KDDTest+ dataset. Also, the results of ILSTM were compared with conventional LSTM and other optimized versions of LSTM (i.e., LSTM-BOA, and LSTM-CBOA). All four algorithms (LSTM, LSTM-BOA, LSTM-CBOA, and ILSTM) were implemented and executed under the same environment to ensure a fair comparison. The results of these experiments are summarized in Table 10. In addition, the confusion matrix for all implemented algorithms is shown in Fig 15. From these results, it can be noticed that ILSTM outperformed almost all other algorithms under all evaluation metrics. This is due to the integration of CBOA with PSO in the proposed ILSTM algorithm.

**Table 10. Comparison between LSTM, LSTM-BOA, LSTM-CBOA and ILSTM using KDDTest+ in Multi-class classification with average 10 runs.**

| Method | Class | DR | Prec | SPC | FAR | FNR | F-Score | MCC | KAPPA |
|---|---|---|---|---|---|---|---|---|---|
| LSTM | Normal | 91.38 | 74.28 | 76.05 | 23.95 | 8.62 | 81.95 | 66.90 | 16.86 |
| | Dos | 84.48 | 95.38 | 97.98 | **2.02** | 15.52 | 89.60 | 85.25 | 40.34 |
| | Prob | 82.45 | 69.72 | 95.69 | 4.31 | 17.56 | 75.55 | 72.66 | 77.28 |
| | R2L | 25.99 | 88.40 | 99.53 | 0.48 | 74.00 | 40.18 | 44.91 | **84.58** |
| | U2R | 19.80 | 12.5 | 98.75 | 1.25 | 80.20 | 15.33 | 14.78 | 97.71 |
| LSTM-BOA | Normal | 95.19 | 76.52 | 77.89 | 22.11 | 4.81 | 84.84 | 72.57 | 14.41 |
| | Dos | 88.29 | 89.23 | 94.73 | 5.27 | 11.71 | 88.76 | 83.25 | 37.64 |
| | Prob | 73.61 | **87.91** | **98.78** | **1.22** | 26.39 | 80.13 | 78.35 | 80.67 |
| | R2L | 20.59 | **94.50** | 99.83 | 0.17 | 79.41 | 33.81 | 41.55 | **85.41** |
| | U2R | 22.28 | 9.83 | 98.15 | 1.85 | 77.72 | 13.64 | 13.65 | 97.10 |
| LSTM-CBOA | Normal | 90.63 | 77.44 | 80.13 | 19.87 | 10.30 | 83.09 | 69.20 | 18.53 |
| | Dos | 89.22 | 89.16 | 94.69 | 5.32 | 11.83 | 88.63 | 83.11 | 37.66 |
| | Prob | 80.917 | 72.00 | 96.09 | 3.91 | 17.20 | 76.98 | 74.24 | 77.54 |
| | R2L | 28.48 | 91.76 | 99.59 | 0.41 | 69.61 | 45.44 | 49.71 | 84.12 |
| | U2R | 17.871 | 22.10 | 99.26 | 0.74 | 83.42 | 16.81 | 17.33 | 98.23 |
| ILSTM | Normal | **95.57** | **88.53** | **91.42** | **8.80** | **4.08** | 91.06 | 84.63 | **19.54** |
| | Dos | **92.23** | **95.45** | 97.93 | 2.08 | **4.59** | 94.28 | 91.46 | 42.34 |
| | Prob | **86.55** | 82.66 | 98.36 | 1.87 | **9.75** | 84.62 | 84.57 | 79.73 |
| | R2L | **67.13** | 89.77 | **99.53** | 0.41 | 23.95 | 78 | 75.89 | 83.91 |
| | U2R | **25.10** | 45.18 | 99.89 | 0.12 | 72.28 | 26.67 | 24.82 | 98.95 |

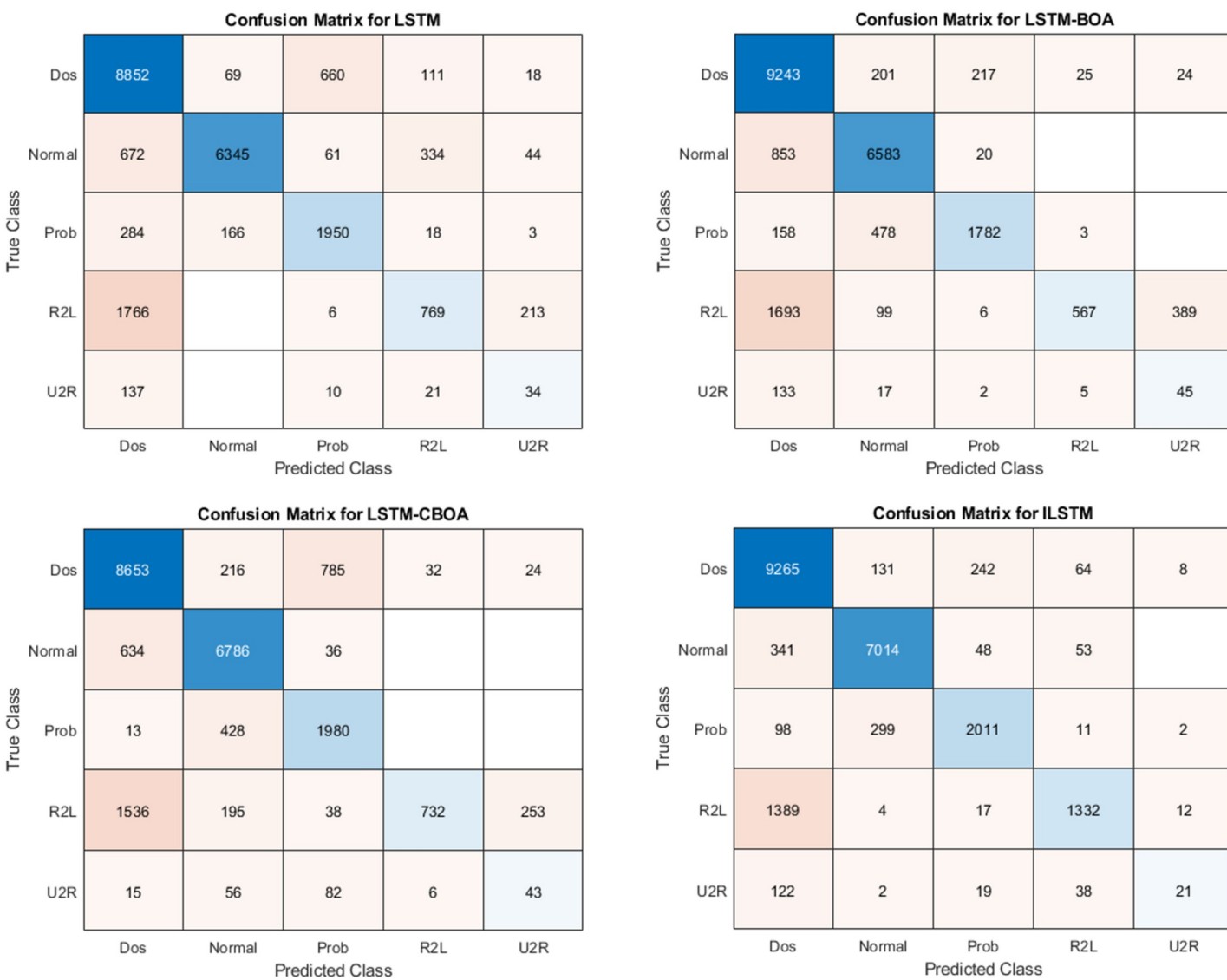

**Fig 15. Confusion matrices of ILSTM using KDDTest+ for multi-classification.**

Also, the ILSTM was compared with the LSTM in terms of the number of iterations needed to achieve the highest accuracy and the results were plotted in Fig 16. From this figure, it can be seen that the proposed ILSTM algorithm achieved a higher accuracy with fewer iterations compared with the conventional LSTM, which reaches 79.72% with 100 iterations as shown in Fig 16(a); however, the ILSTM only needs 72 iterations to attain an accuracy of 88.17% as shown in Fig 16(b). So, the proposed ILSTM can improve intrusion detection performance and also save on computational costs.

**6.2.2 ILSTM performance using KDDTest-21.** To further evaluate the proposed ILSTM, we repeated the same experiment above but using a different dataset, namely KDDTest-21. The results of this experiment are summarized in Table 11. Also, confusion matrices of all compared algorithms in this experiment are plotted in Fig 17. From this table and the

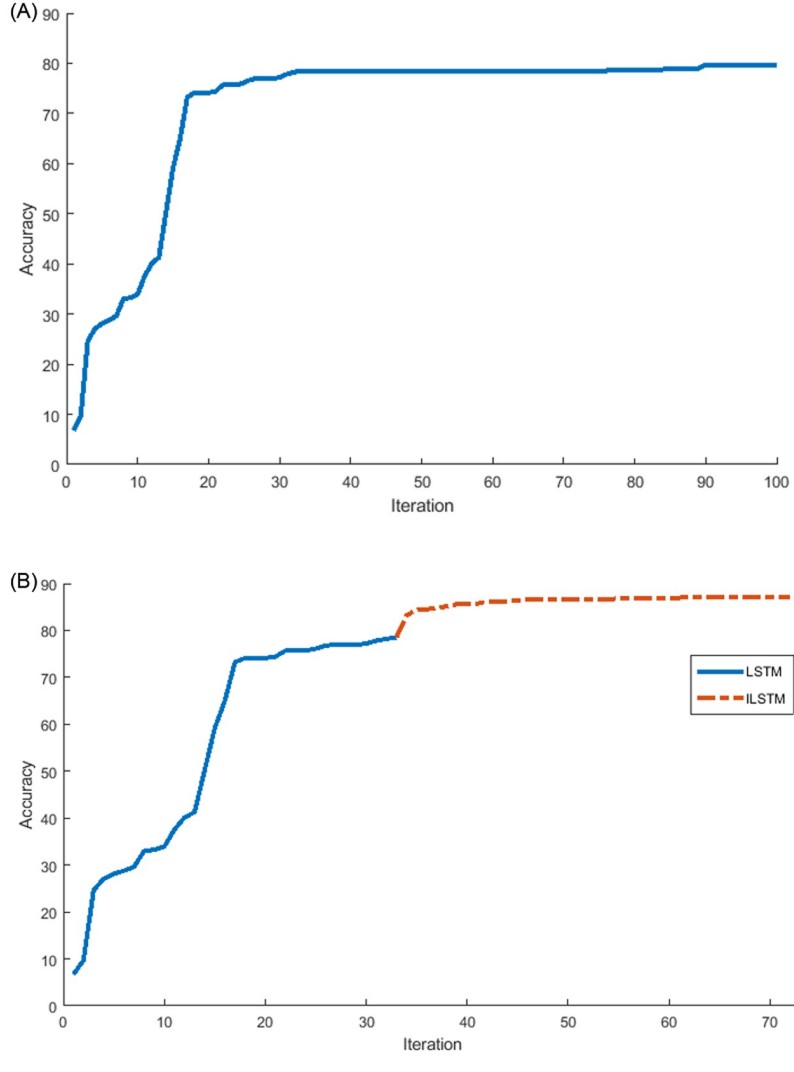

**Fig 16. LSTM vs ILSTM for KDDTest+ in multi-class classification.** A: LSTM. B: ILSTM.

confusion matrix, it can be seen that the ILSTM achieved the best results when compared to other implemented algorithms.

Under the multi-classification scenario, we also investigated the relationship between the accuracy of ILSTM and its number of iterations in comparison with the original LSTM. The results of this experiment were plotted in Fig 18. From this figure, it can be noticed that the optimization of LSTM using CBOA and PSO can boost the accuracy by 20% in 10 iterations, while the accuracy of conventional LSTM remains constant starting from iteration 60 to iteration 100. It can be seen that the conventional LSTM gave an accuracy of 52.54% with 100 iterations as illustrated in Fig 18(a), while ILSTM achieved an accuracy of 76.73% in the same number of iterations as illustrated in Fig 18(b), where optimization process started just after the LSTM accuracy becomes constant, i.e., at iteration 60.

**6.2.3 Efficiency of ILSTM algorithm in multi-class classification.** For a further thorough evaluation of the optimization of LSTM, the MSE was computed using Eq 34 in the context of multi-class classification. The results are summarized in Fig 19 which shows

**Table 11. Comparison between LSTM, LSTM-BOA, LSTM-CBOA and ILSTM using KDDTes-21 in multi-class classification with average 10 runs.**

| Method | Class | DR | Prec | SPC | FAR | FNR | F-Score | MCC | KAPPA |
|---|---|---|---|---|---|---|---|---|---|
| LSTM | Normal | 52.84 | 25.82 | 66.32 | 33.68 | 47.17 | 34.69 | 15.28 | 53.74 |
| | Dos | 60.69 | 74.71 | 88.12 | 11.88 | 39.31 | 66.97 | 51.43 | 42.66 |
| | Prob | 77.81 | 74.31 | 93.16 | 6.84 | 22.19 | 76.02 | 69.78 | 60.95 |
| | R2L | 29.376 | **95.74** | **99.60** | **0.40** | 70.63 | 44.96 | 47.56 | **70.96** |
| | U2R | **23.5** | 8.39 | 95.60 | 4.40 | 76.5 | **12.37** | 11.59 | 93.72 |
| LSTM-BOA | Normal | 45.59 | 40.32 | 85.03 | 14.97 | 54.41 | 42.79 | 29.22 | 65.95 |
| | Dos | 79.80 | 72.13 | 82.17 | 17.83 | 20.20 | 75.77 | 60.81 | 32.91 |
| | Prob | 68.65 | **84.96** | **96.91** | 3.09 | 31.35 | 75.94 | 71.22 | 65.14 |
| | R2L | 63.47 | 75.44 | 93.75 | 6.26 | 36.53 | **68.94** | 60.93 | 60.46 |
| | U2R | 18.00 | 10.14 | 97.26 | 2.74 | 82.00 | 12.97 | 11.53 | 95.40 |
| LSTM-CBOA | Normal | 43.83 | 48.35 | 88.89 | 11.11 | 57.50 | 44.24 | 33.18 | 68.69 |
| | Dos | 83.64 | 70.98 | 76.95 | 20.37 | 16.36 | 74.90 | 59.61 | 29.42 |
| | Prob | 81.05 | 80.50 | 94.64 | 5.36 | 19.16 | 80.00 | 75.06 | 61.04 |
| | R2L | 50.02 | 84.66 | 96.46 | 3.54 | 53.7 | 56.87 | 53.75 | 65.67 |
| | U2R | 18.10 | 8.45 | 96.44 | 3.56 | 81.9 | 11.09 | 9.86 | 94.62 |
| ILSTM | Normal | **59.81** | **67.70** | **93.44** | **6.56** | **37.23** | **60.61** | **52.92** | **70.52** |
| | Dos | **91.83** | **82.13** | **88.39** | **11.61** | **6.71** | **84.80** | **75.93** | 30.91 |
| | Prob | **86.26** | 82.68 | 95.95 | **4.05** | 14.62 | 81.32 | 76.62 | 62.59 |
| | R2L | **64.87** | 87.08 | 98.93 | 1.07 | **35.05** | 67.75 | **61.50** | 70.30 |
| | U2R | 17.9 | **28.95** | **99.05** | **0.95** | 82.10 | **12.05** | 11.93 | **97.31** |

that optimizing the weights of a conventional LSTM network can enhance the multi-class accuracy (i.e., detecting different type of attacks) while also achieving the lowest MSE in multi-class classification for the KDDTest+ and KDDTest-21 datasets, where the KDDTest+ in Fig 19(a) provided the lowest MSE compared with the KDDTest-21 in Fig 19(b).

**6.2.4 Wilcoxon signed-rank test for multi-class classification on KDDTest+ and KDDTest-21 datasets.** To evaluate the significance of the ILSTM, we implemented the Wilcoxon signed-rank test in multi-class classification using two testing datasets, KDDTest+ and KDDTest-21 datasets. The Wilcoxon statistics results of the accuracy of the proposed ILSTM algorithm and the conventional LSTM are shown in Table 12. From this table, it can be seen that the value of z is 2.8031 and the p-value is.00512 for the KDDTest + and KDDTest-21 datasets, and the mean difference for KDDTest+ is -7.99 and the mean difference for KDDTest-21 is -21.29. The results also show that the p = 0.00512 is lower than the significance level (0.05). This means that the null hypothesis should be rejected and it means that there is a significant difference between the proposed ILSTM algorithm and the other algorithms. This statistical analysis confirms the numerical results reported above.

As a conclusion of the results given in Tables 10 and 11, ILSTM achieved better than LSTM, LSTM-BOA, and LSTM-CBOA in terms of DR, Spec, FNR, and MCC for all attack classes (Normal, Dos, Prob, R2l, U2R). This means using a hybrid optimization of CBOA and PSO helped in increasing search space for best solutions and finding global optimal solutions in all testing datasets.

**6.2.5 Comparison of the proposed ILSTM algorithm and the other related algorithms.** As in the case of binary classification, we also compared ILSTM with other published

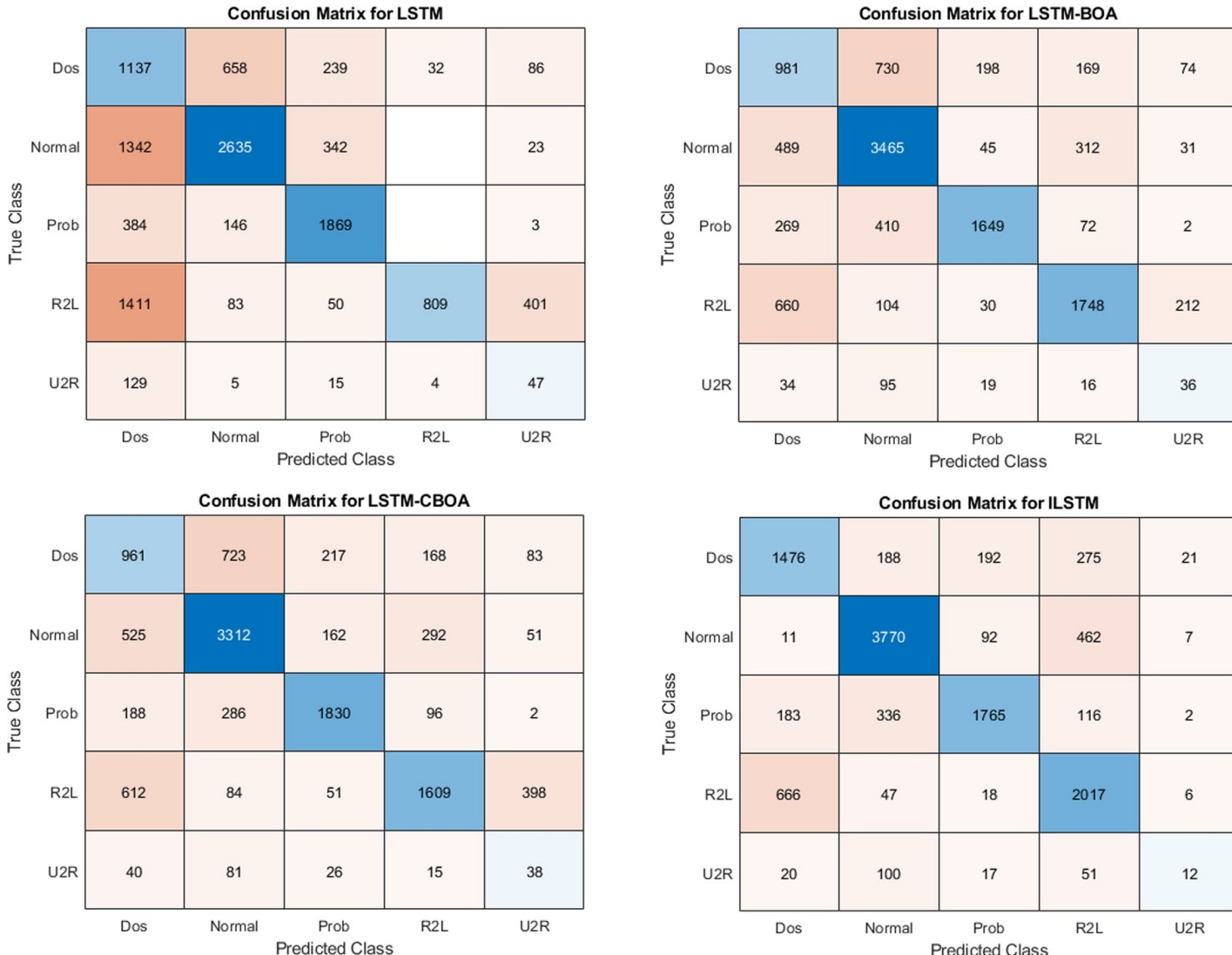

**Fig 17. Confusion matrices for KDDTest-21 in multi classification.**

work about multi-classification attacks. This comparison included deep and machine learning methods which have been published in the literature of intrusion detection context [13, 14, 25]. As shown in Figs 20 and 21, the proposed ILSTM algorithm outperformed other existing machine learning and deep learning methods.

Given the comparison summarized in Table 13, it could be noticed that the proposed ILSTM algorithm achieved the lowest false alarm rate in all types of attacks. Also, the ILSTM can reach higher DR, Precision and f-measure in most types of attacks, the best results are written in bold text. In comparison with other methods, it can be seen that ILSTM gave superior results in Dos, Prob and R2l attacks in terms of Recall, Precision, F-score and FAR. Additionally, ILSTM can produce good results in normal but not as well under R2L attacks. This may be due to the limited number of instants in U2R and R2L attacks in the datasets. Hybrid sampling can help to resolve this issue by achieving results that are as excellent as those of R2L

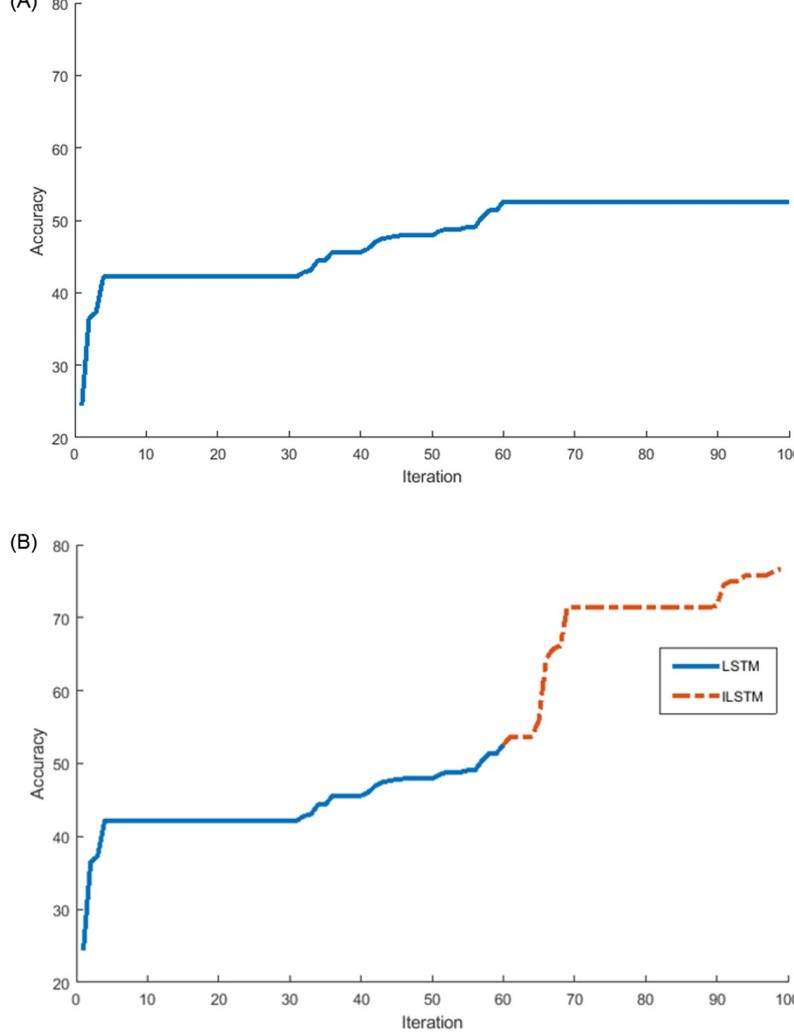

**Fig 18. LSTM vs ILSTM for KDDTest-21 in multi-class classification.** A: LSTM. B: ILSTM.

attacks. The best FAR results for U2R attacks, while also increasing accuracy to 31.88, but still not the best in most metrics for U2R attacks.

## 6.3 Experiment 3: ILSTM performance for binary classification on LITNET-2020 dataset

Similarly to NSLKDD dataset, the ILSTM algorithm was evaluated on the LITNET-2020 dataset, described earlier. Also, the same nine performance metrics were used for evaluating the conventional LSTM and the proposed ILSTM algorithms. A summary of the results is given in Table 14. The confusion matrix for binary classification of LSTM before applying the optimization is given in Fig 22A). On the other hand, the results of applying optimization (i.e., ILSTM) is given in Fig 22B). Also, Fig 23 shows how the accuracy increased after using the ILSTM which starts the optimization process at iteration 58 to improve the accuracy of the original LSTM which was a constant at iteration 68. As shown in Fig 23B, ILSTM improved the accuracy from 92% to 94% at iteration 84 while the conventional LSTM gave an accuracy

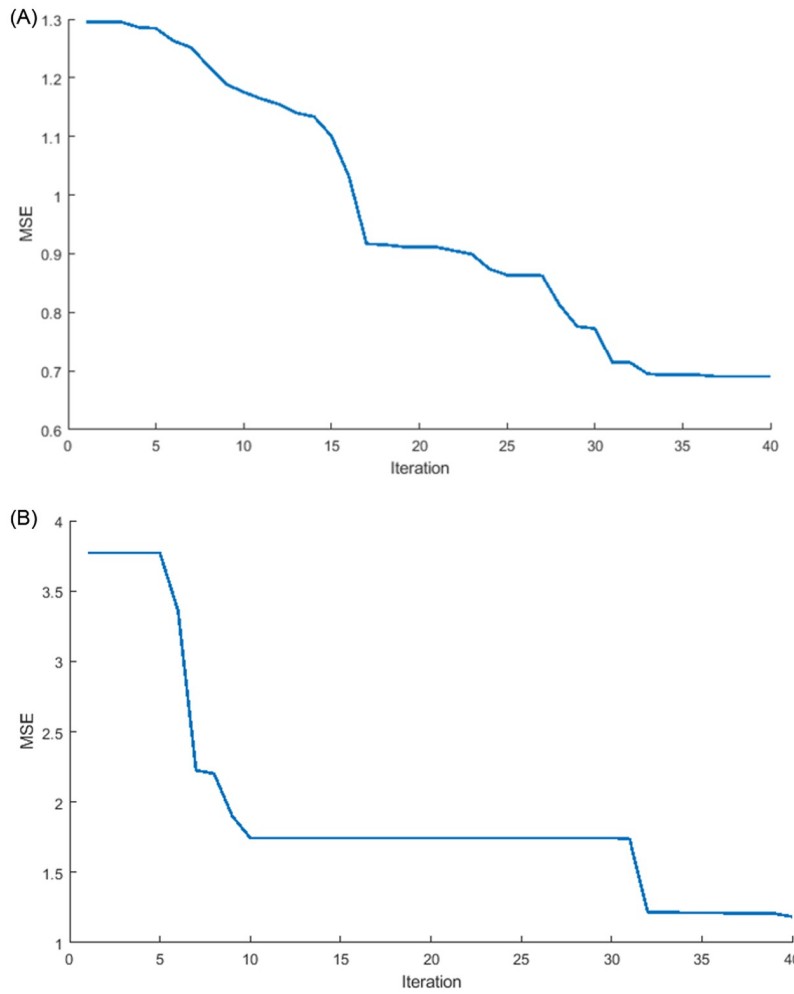

**Fig 19. MSE evaluation for the ILSTM algorithm in multi-class classification.** A: KDDTest+. B: KDDTest-21.

**Table 12. Statistic test for KDDTest+ and KDDTest-21 in multi-class classificatio.**

| KDDTest+ | | | | | | KDDTest-21 | | | | | |
|---|---|---|---|---|---|---|---|---|---|---|---|
| LSTM | ILSTM | Sign | Abs | R | SignR | LSTM | ILSTM | Sign | Abs | R | SignR |
| 76.5 | 87.06 | -1 | 10.56 | 9 | 9 | 45.61 | 76.73 | -1 | 31.12 | 10 | -10 |
| 75.92 | 85.66 | -1 | 9.74 | 7 | 7 | 44.94 | 70.25 | -1 | 25.31 | 7 | -7 |
| 76.88 | 85.82 | -1 | 8.94 | 6 | 6 | 47.16 | 72.51 | -1 | 25.35 | 8 | -8 |
| 77.92 | 88.17 | -1 | 10.25 | 8 | 8 | 49.06 | 67.28 | -1 | 18.22 | 3 | -3 |
| 79.14 | 83.12 | -1 | 3.98 | 1 | 1 | 42.18 | 66.16 | -1 | 23.98 | 4 | -4 |
| 79.51 | 87.54 | -1 | 8.03 | 4 | 4 | 51.86 | 76.29 | -1 | 24.43 | 6 | -6 |
| 79.3 | 85.56 | -1 | 6.26 | 2 | 2 | 54.83 | 65.6 | -1 | 10.77 | 1 | -1 |
| 78.56 | 86.95 | -1 | 8.39 | 5 | 5 | 53.66 | 77.9 | -1 | 24.24 | 5 | -5 |
| 75.5 | 86.81 | -1 | 11.31 | 10 | 10 | 44.96 | 74.51 | -1 | 29.55 | 9 | -9 |
| 77.43 | 84.1 | -1 | 6.67 | 3 | 3 | 55.34 | 69.6 | -1 | 14.26 | 2 | -2 |

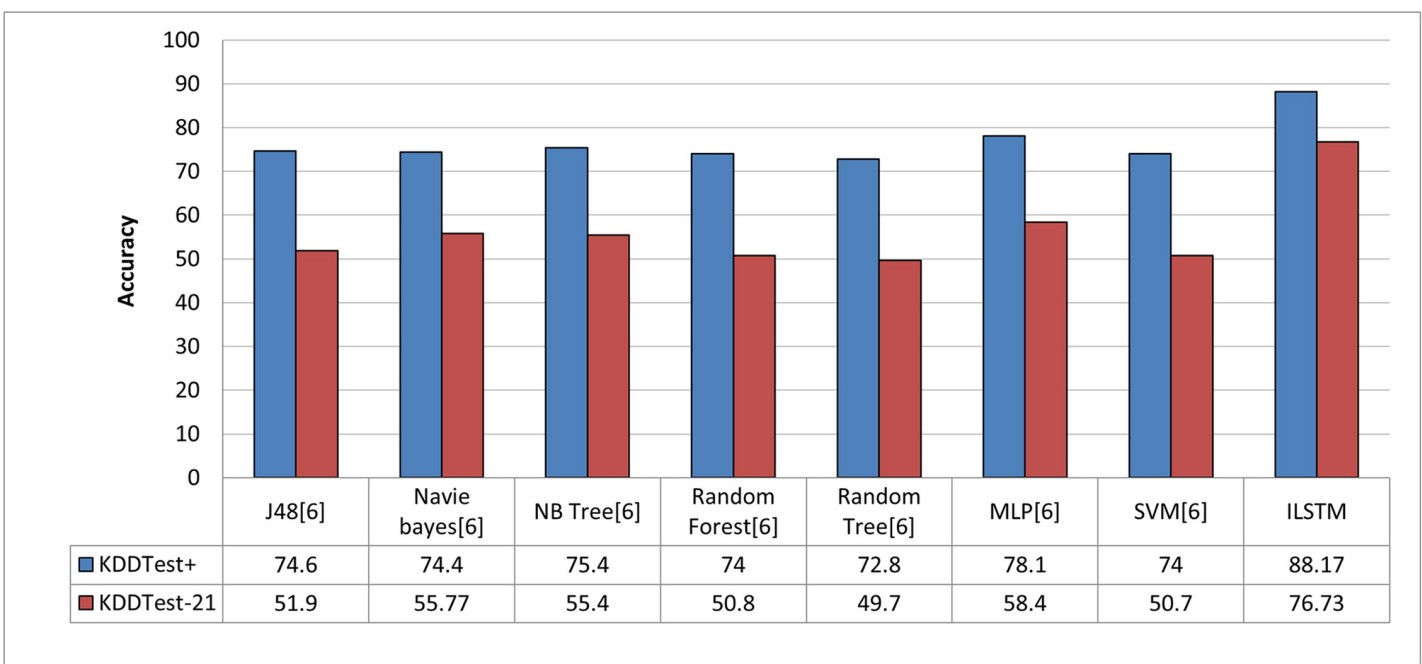

**Fig 20. Comparison with machine learning methods in multi-class classification.**

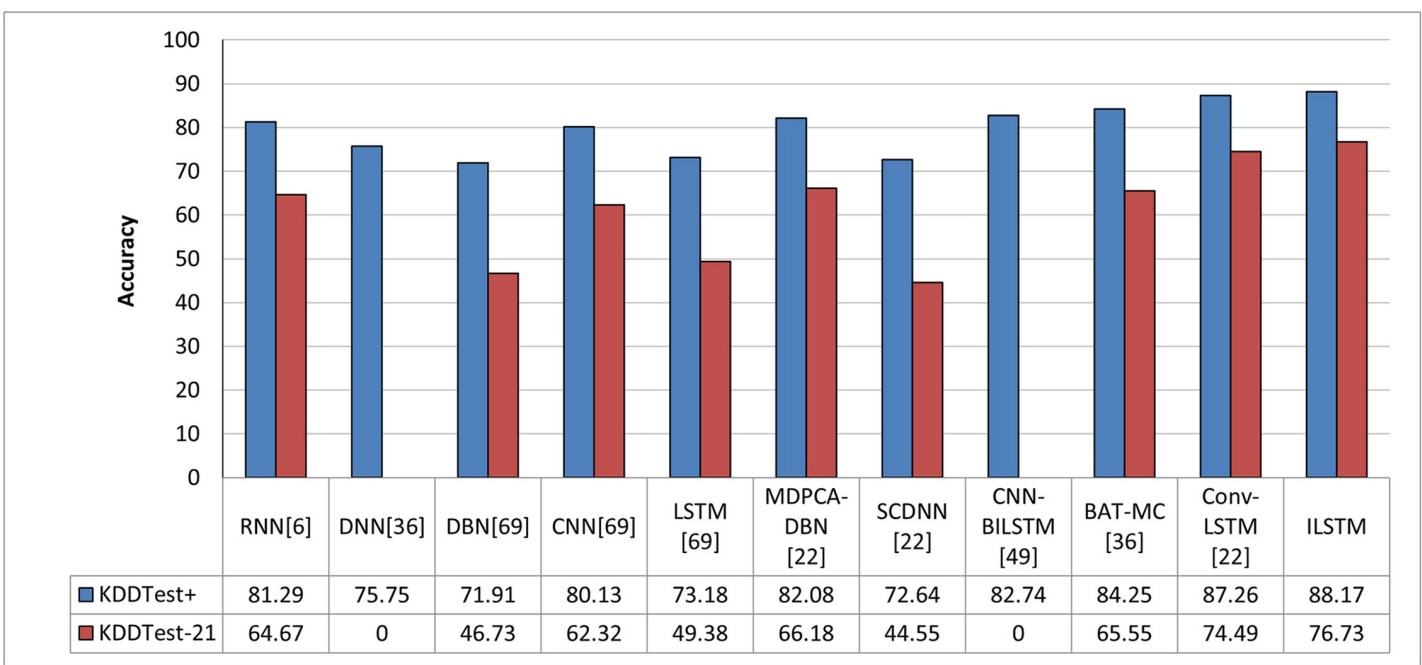

**Fig 21. Comparison with deep learning methods in multi-class classification.**

**Table 13. Comparison between other methods in literature using KDDTest+ for multi-class classification.**

| Method | Class | Recall | Precision | F-Score | FAR |
|---|---|---|---|---|---|
| RNN [14] | Normal | 97 | 73 | 83 | 22.50 |
| | Dos | 83.49 | 96 | 89 | 2.06 |
| | Prob | 83.40 | 85 | 84 | 2.16 |
| | R2L | 24.69 | 83 | 38 | 0.80 |
| | U2R | 11.50 | 66 | 20 | 0.07 |
| CNN [60] | Normal | 96.73 | NA | NA | 29.50 |
| | Dos | 85.26 | NA | NA | 1.92 |
| | Prob | 73.97 | NA | NA | 1.71 |
| | R2L | 17.78 | NA | NA | 0.27 |
| | U2R | 8.95 | NA | NA | **0.004** |
| DSN [42] | Normal | 97.32 | NA | NA | NA |
| | Dos | 90.7 | NA | NA | NA |
| | Prob | 90.08 | NA | NA | NA |
| | R2L | 49.02 | NA | NA | NA |
| | U2R | 18 | NA | NA | NA |
| BAT-MC [25] | Normal | **97.50** | NA | NA | 25.70 |
| | Dos | 87.55 | NA | NA | 1.52 |
| | Prob | 85.76 | NA | NA | 1.15 |
| | R2L | 44.25 | NA | NA | 0.91 |
| | U2R | 20.95 | NA | NA | 0.09 |
| CNN-BLSTM [26] | Normal | 94.11 | 86.77 | 90.29 | NA |
| | Dos | 85.24 | **96.21** | 90.39 | NA |
| | Prob | 68.56 | 64.86 | 66.66 | NA |
| | R2L | 60.45 | 61.32 | 60.21 | NA |
| | U2R | **58.95** | **61.32** | **60.11** | NA |
| ILSTM | Normal | 96.14 | **91.90** | **92.55** | **6.21** |
| | Dos | **97.60** | **96.12** | **94.82** | **1.56** |
| | Prob | **90.21** | **86.05** | **85.48** | **1.13** |
| | R2L | **87.07** | **91.29** | **88** | **0.016** |
| | U2R | 31.88 | 57.14 | 33.6 | 0.053 |

of 92.06% after 100 iterations as shown in Fig 23A. From Fig 23, it can be noticed that ILSTM has improved DR, Where ILSTM achieved 92.55% for DR but LSTM gave 87.46%.

## 6.4 Experiment 4: ILSTM performance for mulit-class classification on LITNET-2020 dataset

In this experiment, the proposed ILSTM is also compared with the original LSTM for a multi-classification scenario. In this experiment, the nine performance metrics were employed in

**Table 14. Comparison between LSTM and ILSTM using LITNET-2020 in binary classification.**

| Method | ACC | DR | SPC | Preci | FAR | FNR | F1-Score | MCC | KAPPA |
|---|---|---|---|---|---|---|---|---|---|
| LSTM | 92.06 | 87.46 | 94.17 | 87.31 | 5.83 | 12.54 | 87.38 | 81.59 | 81.59 |
| ILSTM | 93.97 | 92.55 | 94.59 | 88.61 | 5.41 | 7.45 | 90.61 | 86.22 | 86.18 |

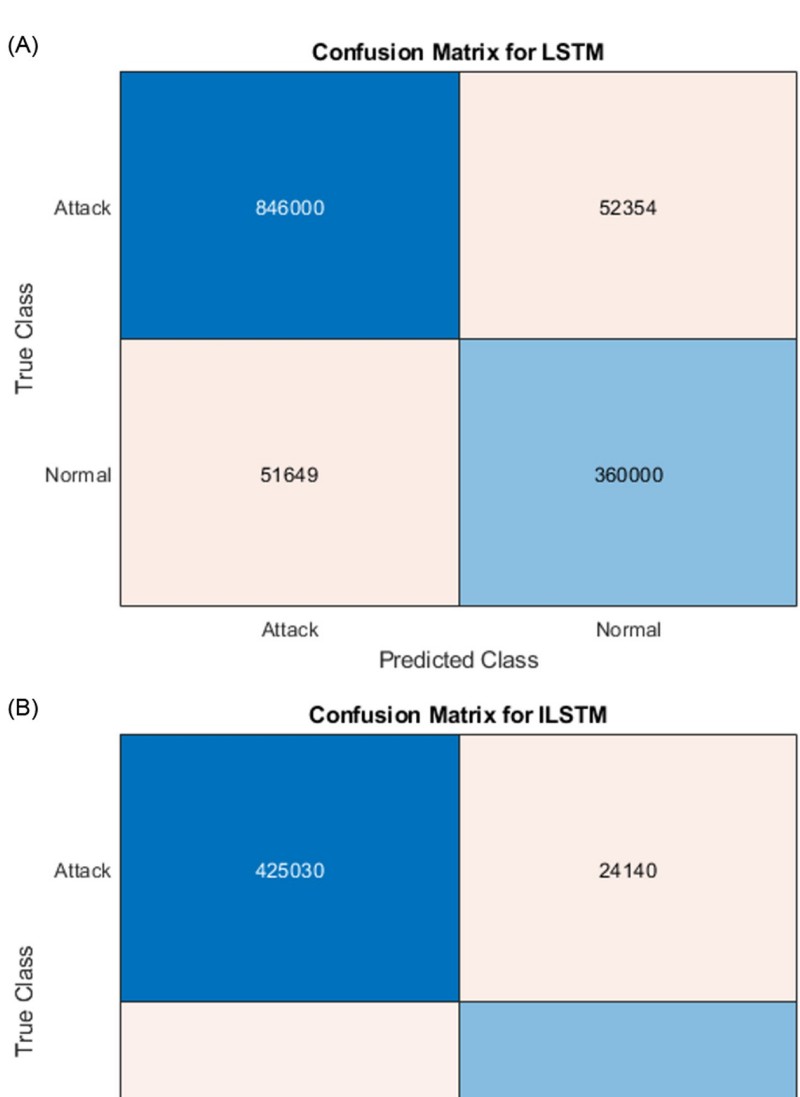

**Fig 22. Confusion matrices for LITNET-2020 dataset in binary classification.** A: LSTM. B: ILSTM.

this comparison and the results are summarised in Table 15. The bold text in this table indicates the best values of the results for the thirteen attack classes. These results showed that ILSTM can increase DR for the "fragmentation" attack, from 0% to 28.37% and also improved the DR for spam attack from 66.23% to 100%. Also, the confusion matrix of the multi-classification attack detection for the LSTM and ILSTM algorithms is given in Figs 24 and 25, respectively.

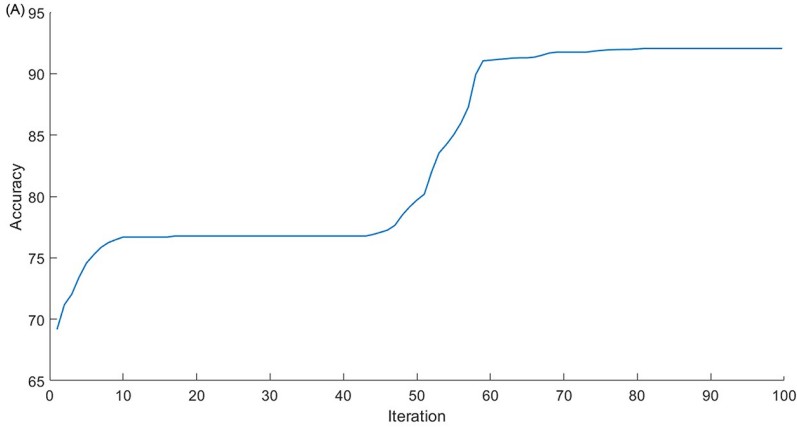

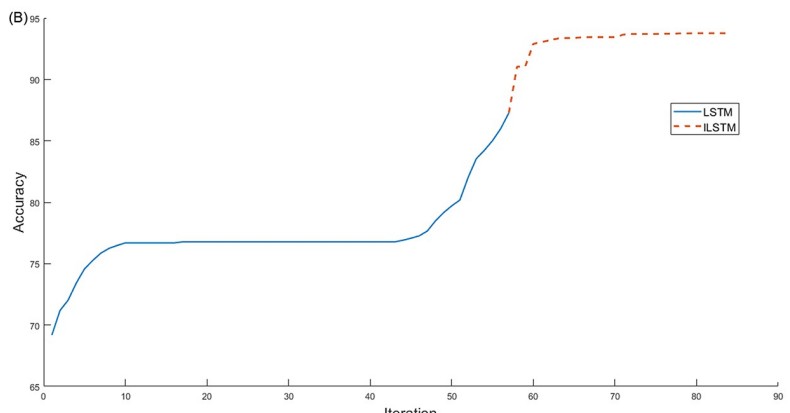

**Fig 23. LSTM vs ILSTM for LITNET-2020 in binary classification.** A: LSTM. B: ILSTM.

Using the LITNET-2020, we also investigated the relationship between the accuracy of ILSTM and the required number of iterations and compared it with the original LSTM. The results of this experiment were plotted in Fig 26. This figure shows that the proposed ILSTM algorithm achieved a higher accuracy rate (i.e., 95.77%) compared with the accuracy of the conventional LSTM, i.e., 91.04%. ILSTM results were achieved using 90 iterations after which the accuracy value became constant while LSTM achieved it accuracy results using 85 iterations after which the accuracy became constant.

## 7 Conclusion and future work

In this paper, we developed an improved version of LSTM (called ILSTM) to improve the accuracy of LSTM-based intrusion detection system. The ILSTM made use of a combination of two swarm optimisation algorithms, CBOA and PSO, to determine the best weights for the LSTM network. The ILSTM consists of two phases: one for training the deeper LSTM network with the best parameters to get initial weights and another for optimizing these weights using CBOA and PSO. A comprehensive evaluation was conducted to assess the efficiecy of the proposed ILSTM algorithm for intrusion detection systems. Two public datasets (NSL-KDD and LITNET-2020) and nine evaluation metrics were used. The results showed that the proposed

**Table 15. Comparison between LSTM, ILSTM using LITNET-2020 in multi-class classification.**

| Method | Class | DR | Prec | SPC | FAR | FNR | F-Score | MCC | KAPPA |
|---|---|---|---|---|---|---|---|---|---|
| LSTM | Benign | 100 | 100 | 100 | 0 | 0 | 100 | 100 | 44.78 |
| | SYN flood | 99.99 | 99.94 | 99.99 | 0.01 | 0.01 | 99.97 | 99.96 | 60.06 |
| | Code red | 99.76 | 94.58 | 99.20 | 0.80 | 0.24 | 97.10 | 96.73 | 74.79 |
| | UDP flood | 100 | 91.92 | 99.32 | 0.68 | 0 | 95.79 | 95.55 | 85.11 |
| | Smurf | 98.90 | 75.64 | 96.81 | 3.19 | 1.10 | 85.72 | 85.02 | 79.32 |
| | LAND DoS | 78.65 | 93.74 | 99.78 | 0.22 | 21.35 | 85.53 | 85.34 | 92.67 |
| | W32.Blaster | 99.78 | 60.33 | 97.47 | 2.53 | 0.22 | 75.19 | 76.59 | 90.24 |
| | HTTP flood | 12.89 | 45.52 | 99.72 | **0.28** | 87.11 | 20.09 | 23.56 | 97.76 |
| | ICMP flood | 78.63 | 98.82 | 99.89 | 0.11 | 21.37 | 87.58 | 86.97 | **81.07** |
| | Port scan | 0 | 0 | 100 | 0 | 100 | NaN | 0.01 | 99.52 |
| | Reaperworm | **0.77** | **1.73** | 99.36 | **0.64** | 99.23 | 1.07 | 0.20 | 97.94 |
| | Spam | 66.23 | **99.75** | 100 | **0** | 33.77 | 79.61 | 81.15 | **98.48** |
| | Fragmentation | 0 | 0 | 100 | 0 | 100 | NaN | 0.01 | **99.13** |
| ILSTM | Benign | 100 | 100 | 100 | 0 | 0 | 100 | 100 | 44.78 |
| | SYN flood | 100 | 99.94 | 100 | 0.01 | 0.02 | 100 | 100 | 60.06 |
| | Code red | 99.58 | **99.88** | **100** | **0.02** | 0 | **99.72** | 100 | **75.41** |
| | UDP flood | 100 | **97.52** | 100 | **0.20** | 0 | **98.74** | **98.66** | **85.52** |
| | Smurf | 100 | **94.41** | 100 | **0.59** | **0** | **97.12** | **96.88** | **81.33** |
| | LAND DoS | 78.65 | **98.68** | 100 | 0.04 | 21 | **87.54** | **87.68** | **92.83** |
| | W32.Blaster | 100 | **70.89** | **98.41** | 1.58 | **0** | 82.9 | 83.45 | **91.12** |
| | HTTP flood | **37.94** | **54.94** | 99.44 | 0.55 | **62** | **44.88** | **44.86** | 97.06 |
| | ICMP flood | **100** | **100** | 100 | **0** | **0** | **100** | **100** | 78.66 |
| | Port scan | **72.03** | 100 | **100** | 0 | **28** | 83.74 | 84.81 | 99.18 |
| | Reaperworm | 0 | 0 | 99.06 | 0.93 | 100 | NaN | **1.17** | 97.66 |
| | Spam | **100** | 68.02 | 99.56 | 0.43 | **0** | 80.97 | 82.30 | 97.75 |
| | Fragmentation | **28.37** | **100** | 100 | 0 | **72** | **44.19** | **53.10** | 98.88 |

ILSTM algorithm is better than the orginal LSTM and two optimized versions of it (i.e., LSTM-PSO, and LSTM-CBOA) in two main cases: binary and multi-class classification. These results were also achieved using a few number of iterations and these were supported by confusion matrices for all the implemented algorithms. Additionally, by comparing the proposed

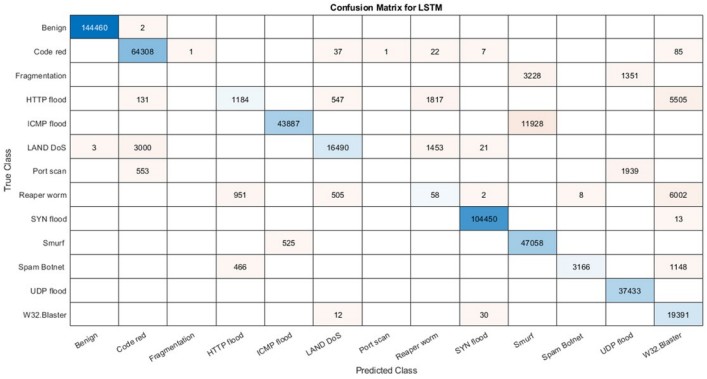

**Fig 24. Confusion matrix for LITNET-2020 dataset using LSTM algorithm.**

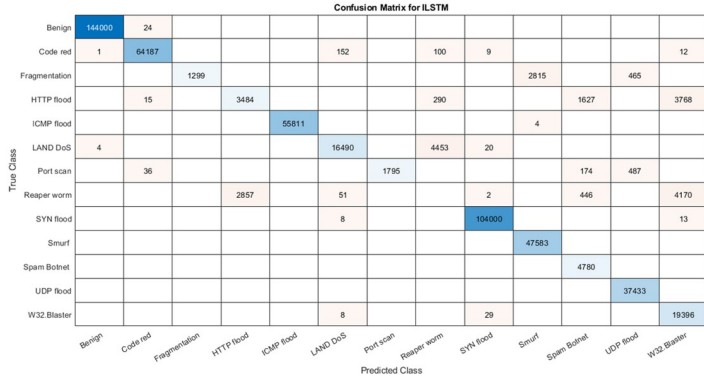

**Fig 25. Confusion matrix for LITNET-2020 dataset using ILSTM algorithm.**

ILSTM algorithm with published and related machine and deep learning methods, the ILSTM yields superior results in terms of accuracy, detection rate, precision, and f-measure when testing on KDDTest+ and KDDTest-21. It was noticed that our proposed algorithm accomplished excellent results when applying optimization, but it needs more time to optimize the population within large datasets. So in future work, it is planned to apply optimization with a faster

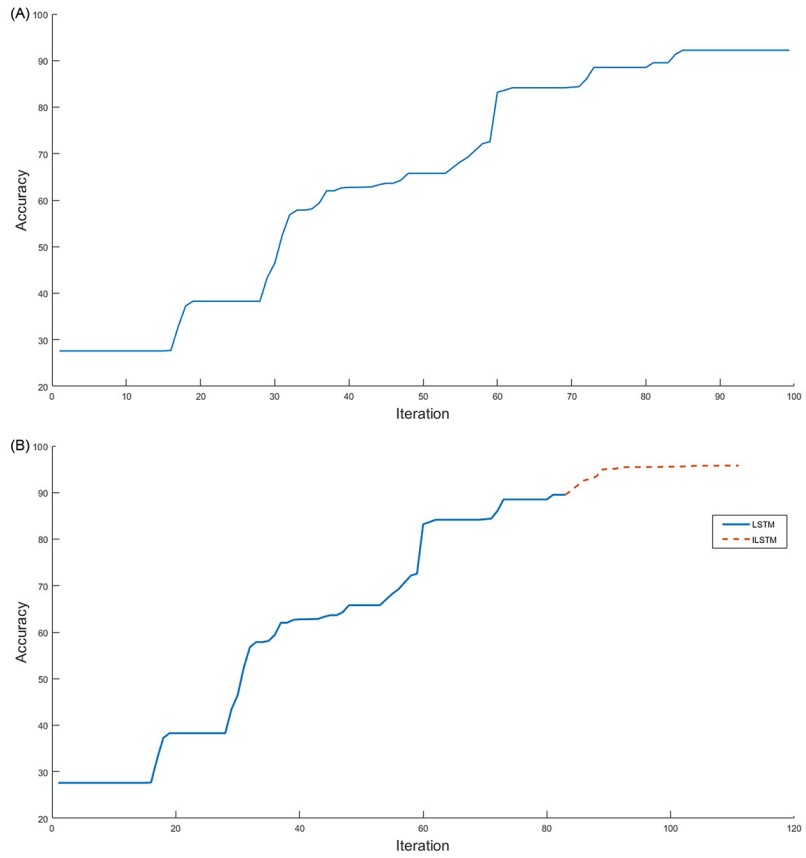

**Fig 26. LSTM vs ILSTM for LITNET-2020 in multi-class classification.** A: LSTM. B: ILSTM.

algorithm. The limitations of the study are as follows: (1) problem of optimization size and computational effort where the proposed algorithm can take a long time when applying it on big data with millions of instances, as in the case of the LITNET-2020 dataset. (2) computational resources: if the problem size is too large, it might not be possible to store the processing data in the memory of the computer running this algorithm.

## Author Contributions

**Conceptualization:** Ahmed Fouad Ali, Tarek Gaber.

**Data curation:** Asmaa Ahmed Awad.

**Methodology:** Asmaa Ahmed Awad, Ahmed Fouad Ali, Tarek Gaber.

**Resources:** Ahmed Fouad Ali.

**Software:** Asmaa Ahmed Awad.

**Supervision:** Ahmed Fouad Ali, Tarek Gaber.

**Validation:** Tarek Gaber.

**Visualization:** Asmaa Ahmed Awad.

**Writing – original draft:** Asmaa Ahmed Awad.

**Writing – review & editing:** Ahmed Fouad Ali, Tarek Gaber.

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
