## [Decision Letter · Decision Letter 0]

5 Jan 2023

PONE-D-22-34491An Improved Long Short Term Memory Network For Intrusion DetectionPLOS ONE

Dear Dr. Gaber,

Thank you for submitting your manuscript to PLOS ONE. After careful consideration, we feel that it has merit but does not fully meet PLOS ONE’s publication criteria as it currently stands. Therefore, we invite you to submit a revised version of the manuscript that addresses the points raised during the review process.

We look forward to receiving your revised manuscript.

Kind regards,

Nebojsa Bacanin

Academic Editor

PLOS ONE

Additional Editor Comment:

Dear Authors,

please revise proposed manuscript carefully according to all reviewers' comments.

All the best with your submission.

Reviewers' comments:

Reviewer's Responses to Questions

**Comments to the Author**

1. Is the manuscript technically sound, and do the data support the conclusions?

Reviewer #1: Yes

Reviewer #2: Yes

Reviewer #3: Yes

Reviewer #4: Yes

Reviewer #5: Yes

Reviewer #6: Yes

Reviewer #7: Yes

2. Has the statistical analysis been performed appropriately and rigorously? 

Reviewer #1: N/A

Reviewer #2: Yes

Reviewer #3: N/A

Reviewer #4: N/A

Reviewer #5: Yes

Reviewer #6: Yes

Reviewer #7: No

3. Have the authors made all data underlying the findings in their manuscript fully available?

Reviewer #1: Yes

Reviewer #2: Yes

Reviewer #3: Yes

Reviewer #4: Yes

Reviewer #5: Yes

Reviewer #6: Yes

Reviewer #7: Yes

4. Is the manuscript presented in an intelligible fashion and written in standard English?

Reviewer #1: Yes

Reviewer #2: Yes

Reviewer #3: Yes

Reviewer #4: Yes

Reviewer #5: Yes

Reviewer #6: Yes

Reviewer #7: No

5. Review Comments to the Author

Reviewer #1: The article proposed a hybrid of chaotic butterfly optimization and particle swarm optimization (PSO) algorithms for weights optimization of deep learning model (Long Short-Term Memory) to improve the accuracy of LSTM in intrusion detection task. The paper needs to be improved and revised according to the comments and suggestions presented below before it could be considered for publication.

Major Comments:

1. The paper combines some known heuristic optimisation methods (chaotic butterfly optimization and particle swarm optimization) with a well-known deep learning model (LSTM). Is the novelty of the proposed methodology just the combination of existing methods, or more novelty is introduced. The authors need to state clearly.

2. The introduction section presents a historical outlook on the field of intrusion detection and machine learning algorithms. Such approach is more suitable for a textbook. Instead, I suggest to focus on the state-of-the-art by discussing the most recently published research articles. The literature review missed some of the recently published works relevant to the topic of this paper. The authors are encouraged to read and discuss * Threat analysis and distributed denial of service (DDoS) attack recognition in the internet of things (IoT). Electronics, 11(3) (2022). * A modified grey wolf optimization algorithm for an intrusion detection system. Mathematics, 10(6) (2022). * A novel approach for network intrusion detection using multistage deep learning image recognition. Electronics, 10(15) (2021).

3. Description of various optimisation methods used in this study could be improved by presenting their algorithms in pseudocode.

4. The proposed methodology is evaluated on the NSL-KDD 2009 dataset, which is outdated and no longer represents current threats in computer networks. The authors should include evaluation on more recent network intrusion dataset such as LITNET-2020.

5. How did you select hyperparameter values for LSTM (Table 3)? Did you use any optimisation/finetuning?

6. Discuss the limitations of the proposed methodology.

7. Extend and improve conclusions. Discuss implications and recommendations for future research in this field.

Minor comments:

8. Figure 1 is a commonly used image of LSTM cell. The authors should make sure it is an original creation of theirs.

9. Lines 456, 465, 473, 482: references to Figures are missing.

10. Some references such as ref. [15] are not related to the topic and content of the paper. I suggest to remove or replace with a more appropriate reference.

Reviewer #2: The overall manuscript is good and has new results. However, there are some issues that should be revised to improve this paper.

1. I miss a section that outlines the limitations of your ILSTM algorithm and possibilities of extension. Are there any disadvantages or limits of your method?

2. I suggest add a comparative table in ''Literature review'' to contrast your solution in front of related works. You could discuss the relationship between your solution and past literature. You can cite few papers in related areas.

3. Authors need to confirm that all acronyms are defined before being used for first time. Authors need to confirm that all mathematical notations are defined when being used for first time.

4. The conclusion needs improvements towards major claimed contribution. Write some future directions in the conclusion section.

5. Improve the readability of the manuscript in terms of typos mistakes and errors. There are some grammatical errors and awkward phrasings (suggested proofreading the manuscript after addressing all comments to avoid any typo, grammatical, and lingual mistakes and errors).

6. There are some technical errors in the numbering of the figures (Fig ??, pages 7,14,15,16,17,18). This is probably a consequence of the generation of the Latex document.

Reviewer #3: 1. Avoid using acronyms in the abstract.

2. Add more recent papers that deal with swarm intelligence algorithms applied to the tuning of the machine learning models for intrusion detection. Please include:

https://ieeexplore.ieee.org/abstract/document/9936116

https://peerj.com/articles/cs-956/?td=tw

https://www.sciencedirect.com/science/article/pii/S0167404818303936

https://link.springer.com/chapter/10.1007/978-981-19-4831-2_1

3. How did you choose iterative chaotic map, among many other that exist?

4. Why the butterfly optimization algorithm was selected, among many other metaheuristics?

5. Describe the simulation environment and conducted experiments in more details.

6. If possible, provide statistical test results to show if the obtained improvements are statistically significant.

7. SHAP analysis could also be employed, to determine the importance of each feature on the target variable.

8. Conclusion is very short. Mention the experimental results briefly, and indicate directions of the possible future work in this field.

Reviewer #4: The study proposed using a combination of heuristic optimisation with LSTM deep learning model for solving network intrusion detection problem. The results of extensive experiments on a benchmark dataset are presented. The paper is acceptable, provided the authors improve as suggested below.

1. In the introduction section, discuss most relevant issues of current cybersecurity such as Internet-of-Things, smartphone malwares, malicious attacks against AI. Use appropriate references to support examples of applications mentioned in the first paragraph of the section.

2. Overview of related works should be improved. Focus on the state-of-the-art. Please check doi:10.5755/j01.itc.51.1.29595, doi:10.5755/j01.itc.51.4.31818, doi:10.5755/J01.ITC.50.1.25002, doi:10.5755/j01.itc.48.4.24036, doi:10.3390/electronics10111341. Summarize as a table.

3. NSL-KDD 2009 dataset used in this study is old. Consider providing experiments with a more modern database.

4. MSE is relevant for regression tasks. It is usually not used as a performance metric for classification tasks.

5. Explain parameter setting n more detail. How do you select?

6. For experiments, present and discuss the confusion matrices of the classification results.

7. Add more works of other authors to compare with in Table 9.

8. Extend the conclusions section.

Reviewer #5: An Improved Long Short-Term Memory Network For Intrusion Detection

This is an interesting paper and it can be further improved if the following points are addressed.

In the introduction, there should be a few points to be discussed:

· The objectives of the paper.

· The guide to the rest of the sections of the paper.

The literature work did not cover all the new, recent, and powerful algorithms,

Literature work did not cover all the recent algorithms, there is a more recent and powerful algorithm in the recent literature. Authors can get future reading about new optimizer algorithms and advanced variants of LSTM metaheuristics can be found in the link below: https://nci-rc.com/project/

Thus, more results can be obtained if the authors wanted to pursue their research studies.

All equations should be cited if they are not yours.

Some more diagrams or result plots from the software will be more appropriate and appealing to the readers.

Analytical discussion is needed.

The authors can add more future work.

Reviewer #6: The article proposes a hybrid algorithm between deep learning and swarm intelligence. While this is not a first attempt to do such a hybridization, the extensive experimental results appear to be very promising.

There is a relatively good literature review, but still there are some recent relevant articles that should be mentioned, like https://doi.org/10.1007/978-981-19-4831-2_1 or https://doi.org/10.7717/peerj-cs.956.

The representation of the candidate solutions should be explained and exemplified.

The split of the data set between training, validation and test set should be clearly stated. Also, please clarify what set is used (validation or test?) for calculating the accuracy for the fitness function.

The authors were very negligent with the text of the manuscript. There are many typos, many words written in capital letter in the middle of the sentences, words that are several times abbreviated, the figures are not well referenced (they appear with ??).

Reviewer #7: A method to improve a LSTM is proposed and it is tested for network intrusion detection for binary and multi-class classification.

The enumeration of all performance metrics is not necessary in the abstract. There is no need to introduce abbreviations in the abstract if they are not used within the abstract. Also, some abbreviations are introduced several times in the body of the article.

The results look convincing, but, although statistical results are promised in line 72, the statistical significance of the results is not discussed in the experimental results section (btw, the name of the section is “experiment results”, which sounds strange).

The grammar of the article needs to be revised since it is hard to read. It goes from simple mistakes like “it is critical to developing an effective algorithm” in line 143 to more refined ones, like the enumeration in lines 62-74, where the first item starts with “introducing…” (and it does not have a predicate), while the others contain statements and even one paragraph in item 4. Also, another example is in line 258 “In this Phase, We combined”.

6. PLOS authors have the option to publish the peer review history of their article (what does this mean?). If published, this will include your full peer review and any attached files.

Reviewer #1: No

Reviewer #2: **Yes: **Muzafer Saracevic

Reviewer #3: No

Reviewer #4: No

Reviewer #5: **Yes: **okay

Reviewer #6: No

Reviewer #7: No

---

## [Author Response · Author response to Decision Letter 0]

25 Mar 2023

We have made every effort to eliminate all the indicated major and minor inconsistencies and do hope that as a result, the quality of our revised manuscript has improved further. We are uploading three files: 

(a) our point-by-point response to the comments (below) (response to reviewers), a file for each reviewer is attached

(b) an updated manuscript with yellow highlighting indicating changes, and 

(c) a clean updated manuscript without highlights

---

## [Decision Letter · Decision Letter 1]

10 Apr 2023

An Improved Long Short Term Memory Network For Intrusion Detection

PONE-D-22-34491R1

Dear Dr. Gaber,

We’re pleased to inform you that your manuscript has been judged scientifically suitable for publication and will be formally accepted for publication once it meets all outstanding technical requirements.

Kind regards,

Nebojsa Bacanin

Academic Editor

PLOS ONE

Additional Editor Comments (optional):

Dear Authors,

thank you for revising proposed manuscript.

Based on the reviewers' comments and according to my own evaluation, the manuscript can now be accepted.

Warmest regards,

Nebojsa Bacanin

Reviewers' comments:

Reviewer's Responses to Questions

**Comments to the Author**

1. If the authors have adequately addressed your comments raised in a previous round of review and you feel that this manuscript is now acceptable for publication, you may indicate that here to bypass the “Comments to the Author” section, enter your conflict of interest statement in the “Confidential to Editor” section, and submit your "Accept" recommendation.

Reviewer #1: All comments have been addressed

Reviewer #2: All comments have been addressed

Reviewer #7: All comments have been addressed

2. Is the manuscript technically sound, and do the data support the conclusions?

Reviewer #1: Yes

Reviewer #2: Yes

Reviewer #7: Yes

3. Has the statistical analysis been performed appropriately and rigorously? 

Reviewer #1: Yes

Reviewer #2: Yes

Reviewer #7: Yes

4. Have the authors made all data underlying the findings in their manuscript fully available?

Reviewer #1: Yes

Reviewer #2: Yes

Reviewer #7: Yes

5. Is the manuscript presented in an intelligible fashion and written in standard English?

Reviewer #1: Yes

Reviewer #2: Yes

Reviewer #7: Yes

6. Review Comments to the Author

Reviewer #1: The authors have revised well. This valuable manuscript can be accepted for publication in the PLOS.

Reviewer #2: In the revised version of the manuscript, the authors met all the requirements and comments given in the previous review, so I recommend this paper for publishing.

Reviewer #7: (No Response)

7. PLOS authors have the option to publish the peer review history of their article (what does this mean?). If published, this will include your full peer review and any attached files.

Reviewer #1: No

Reviewer #2: No

Reviewer #7: No

---

## [Editor Report · Acceptance letter]

14 Apr 2023

PONE-D-22-34491R1 

An Improved Long Short Term Memory Network For Intrusion Detection 

Dear Dr. Gaber:

I'm pleased to inform you that your manuscript has been deemed suitable for publication in PLOS ONE. Congratulations! Your manuscript is now with our production department. 

Kind regards, 

on behalf of

Dr. Nebojsa Bacanin 

Academic Editor

PLOS ONE